# Single-chain dimers from de novo immunoglobulins as robust scaffolds for multiple binding loops

Jorge Roel-Touris[1,2], Marta Nadal ®[1,2] & Enrique Marcos ®[1] ✉

Antibody derivatives have sought to recapitulate the antigen binding properties of antibodies, but with improved biophysical attributes convenient for therapeutic, diagnostic and research applications. However, their success has been limited by the naturally occurring structure of the immunoglobulin dimer displaying hypervariable binding loops, which is hard to modify by traditional engineering approaches. Here, we devise geometrical principles for de novo designing single-chain immunoglobulin dimers, as a tunable two-domain architecture that optimizes biophysical properties through more favorable dimer interfaces. Guided by these principles, we computationally designed protein scaffolds that were hyperstable, structurally accurate and robust for accommodating multiple functional loops, both individually and in combination, as confirmed through biochemical assays and X-ray crystallography. We showcase the modularity of this architecture by deep-learning-based diversification, opening up the possibility for tailoring the number, positioning, and relative orientation of ligand-binding loops targeting one or two distal epitopes. Our results provide a route to custom-design robust protein scaffolds for harboring multiple functional loops.

The ability of engineered antibodies to bind antigens with high affinity and specificity makes them versatile protein molecules for therapeutic, diagnostic and research applications[1]. Their antigen-binding sites are shaped by a variable region (Fv) assembling two immunoglobulin (Ig) domains, from the heavy and light chains ($V_H$ and $V_L$), each scaffolding three hypervariable loops that can be optimized for binding almost any antigen of interest (Fig. 1a). Despite the vast molecular recognition abilities of conventional antibodies, there has been considerable interest in developing alternative antibody-based formats with superior biophysical and structural properties, including high stability, efficient recombinant expression in microorganisms (for more cost-effective production), access to hidden epitopes, smaller size (for improved tissue penetration) or lower structural complexity (as single polypeptide chains, convenient for display optimization technologies), among others[2-5]. However, a general constraint on traditional engineering strategies for these scaffolds is the naturally occurring β-sandwich structure of the Fv Ig framework, which is hard to modify either at the level of single Ig domains or $V_H$-$V_L$ dimer orientation. For example, single-chain variable fragments[5,6] (scFv) have been engineered to keep the antigen-binding properties of Fvs in a single polypeptide chain by fusing the $V_H$ and $V_L$ Ig domains. Due to the large distance naturally found between the termini of both domains, scFvs require long and flexible linkers, which often compromises stability and structural control, as seen in other fusion constructs combining different protein domains[7]. More complex antibody derivative constructs have been engineered to display bispecific activity, colocalizing proteins of interest. As Fvs display the binding loops only on one side of the scaffold (monovalent), diabodies and other bivalent formats[8-11] have been engineered by concatenating multiple variable domains with linkers designed to properly orient the antigen-binding sites, while disfavoring alternative conformational or oligomerization states, which remains challenging to control. Additionally,

---

[1]Protein Design and Modeling Lab, Department of Structural and Molecular Biology, Molecular Biology Institute of Barcelona (IBMB), CSIC, Baldiri Reixac 10, 08028 Barcelona, Spain. [2]These authors contributed equally: Jorge Roel-Touris, Marta Nadal. ✉e-mail: embcri@ibmb.csic.es

antibody-based formats share the natural $V_H$-$V_L$ orientation where all edge β-strands from the two Ig domains are exposed to the solvent (Fig. 1a, inset), thus increasing aggregation propensity[12,13].

De novo designing Ig domains with tailored structures and dimeric interfaces not tied to natural ones holds promise for small, structurally tunable antibody-like formats with optimal biophysical properties and enabling new mechanisms of action. Two-domain formats have the advantage over single-domain ones to increase the number of anchoring sites for binding loops, opening up the possibility for either enlarging the binding surface to a single epitope (monospecificity) or simultaneously targeting two distal epitopes (bispecificity). Although there has been considerable progress in the design of single-domain β-sheet structures[14], including β-barrel[15,16], jelly-roll[17] and Ig-like[18] folds, the de novo design of single-chain, two-domain Igs anchoring multiple binding loops, as in antibody variable regions, has not been achieved yet. All-β protein domains have been generally harder to design than all-α or mixed α/β folds[19,20], as they are structurally less regular and have a higher risk of misfolding or aggregation; a problem that could be exacerbated when fusing two of such all-β domains into larger single-chain β-sheet structures. Furthermore, de novo proteins have been typically designed with non-functional, short loops for enhanced stability and structural control, but antibody-like formats underlie the challenge of designing structures robust enough for incorporating functional, and often long and flexible, loops at multiple sites, which has received less attention in the field.

Here, we devise principles for de novo designing single-chain Ig dimers with interfaces diverging from Fvs, as a two-domain Ig architecture that is hard to find in nature and optimizes folding stability. We show that single-chain Ig dimeric scaffolds formed by 12 or 14 β-strands can be computationally designed with hyperstability and structural accuracy, as confirmed by X-ray crystal structures. For one of these, we show that β-hairpin sites can efficiently anchor multiple binding loops, both individually and in combination, as computationally predicted and confirmed by biochemical and binding assays. Lastly, we found that the structure of these scaffolds can be highly diversified at the level of domain components, their interfaces, and both number and positioning of functional sites. Overall, we show the robustness, modularity, and versatile potential of de novo single-chain dimeric Ig scaffolds for harboring multiple functional loops.

## Results

### Design principles for edge-to-edge single-chain immunoglobulin dimers

We envisioned that de novo single-chain Ig dimers could be designed through two-layer edge-to-edge dimeric interfaces (Fig. 1b), resulting in extended Ig β-sandwiches. Such interfaces may have three main advantages over the face-to-face one (Fig. 1b), as in the natural $V_H$-$V_L$ interface. First, they harness the strength of edge-to-edge β-sheet interactions, which combine backbone hydrogen-bonding and side-chain packing interactions between paired β-strands. Second, by burying four edge β-strands in the interface, the number of solvent-exposed β-strands gets reduced to a half (i.e., from eight to four), thereby strongly minimizing aggregation propensity. In the face-to-face arrangement, instead, all edge β-strands (eight in total) from the two Ig β-sandwiches get solvent-exposed and hence favor edge-to-edge intermolecular β-sheet interactions[21]—i.e., through backbone hydrogen bonding involving the exposed amine and carbonyl groups—which increases aggregation propensity (preventing such oligomerization-prone interactions is indeed a general challenge when designing β-sheet proteins[18]). And third, the relative orientation between the two Ig domains could be tailored for a stable fusion into a single polypeptide chain favoring folding.

We began by analyzing the structural requirements to optimize folding stability of two-layer, edge-to-edge single-chain dimers of Ig

domains. Two Ig domains could, in principle, be arranged in multiple ways, but we identified three key geometric considerations to achieve a stable fusion through (1) short and rigid connections and (2) minimal structural modifications. First, in Ig domains, the N- and C-terminal strands pack face-to-face on the edge of two β-sheets and are therefore sticky ends available for edge-to-edge strand pairing (Fig. 1c). Depending on the total number of β-strands, both terminal β-strands get oriented in different directions: 7-stranded Ig domains orient both β-strands in the same direction (Fig. 1c, left), while 6- or 8-stranded domains (having one strand less in the C-termini or an extra one in the N-termini, respectively) orient both β-strands in opposite directions (Fig. 1c, right). Second, if the C- and N-terminal strands of the first and second Ig domain, respectively, are paired to each other and connected they form a sequence-local interaction that would favor folding of the single-chain dimer—amino acids close in sequence tend to form contacts in the three-dimensional structure more rapidly[22]. Such local fusion also shortens the distance spanned by the interdomain connection in comparison with alternative orientations (Fig. 1c, bottom), which further minimizes the risk of misfolding due to incorrect formation of strand pairs. Interdomain pairing orientations compatible with a sequence-local fusion are dictated by the total number of strands of each Ig domain. For example, pairs of Ig domains with the same number of strands must have parallel or antiparallel pairing depending on whether such number is odd or even, respectively—alternative arrangements would collapse both domains or require long-range connections between strands not paired to each other (Fig. 1c). Additionally, the interdomain pairing orientation determines the structure of the fusion to be rigid. If the pairing is parallel, the two strands could be bridged with an α-helix connection packing against one β-sheet, forming a well-structured βαβ motif (Fig. 1c, left); while for an antiparallel pairing, both strands could be connected with a short loop, forming a β-hairpin motif (Fig. 1c, right)—design rules on optimal loop lengths and geometries favoring both motifs have been developed[23,24]. Third, the interface between domains is constrained by the requirement of backbone hydrogen bond pairing between two pairs of edge β-strands from four β-sheets simultaneously, and not all pairs of Ig domain structures will be compatible with such arrangement. De novo Ig domains, in contrast to naturally occurring ones[21,25], should better meet such interface requirements as they can be designed with regular backbone geometries on the edge β-strands[18], thus favoring continuous hydrogen bond pairing in the two layers. Also, the two-layer hydrogen bond pairing is likely more optimal if the global β-sandwich geometries of the two Ig domains—in terms of distance separation and rotations between the two β-sheets of each domain—are similar; otherwise, strand pairing in one layer would be incompatible with a second pairing in the other layer (see Supplementary Fig. 1).

### Design and characterization of edge-to-edge, single-chain immunoglobulin dimer structures

We set out to design edge-to-edge, single-chain Ig dimers with parallel and antiparallel orientations. To this end, we started from the crystal structures of two 7-stranded de novo designed Ig monomers[18] (dIG8-CC and dIG14), which showed edge-to-edge homodimeric crystallographic interfaces and are therefore convenient to assess the extent to which local design connections favor folding and stability of more complex β-sheet fusion architectures.

For the parallel orientation, we began with the dIG8-CC crystal structure (PDB 7SKO), which consists of two Ig domains where the N-terminal β-strand of one domain has parallel pairing with the C-terminal β-strand of the other domain. We sought to design a α-helix bridging both β-strands through short and structured βα and αβ loop connections, and packing against one of the two extended β-sheets; overall forming a compact 14-stranded β-sandwich (Fig. 1c, left). We generated protein backbones by Rosetta[26] Monte Carlo fragment

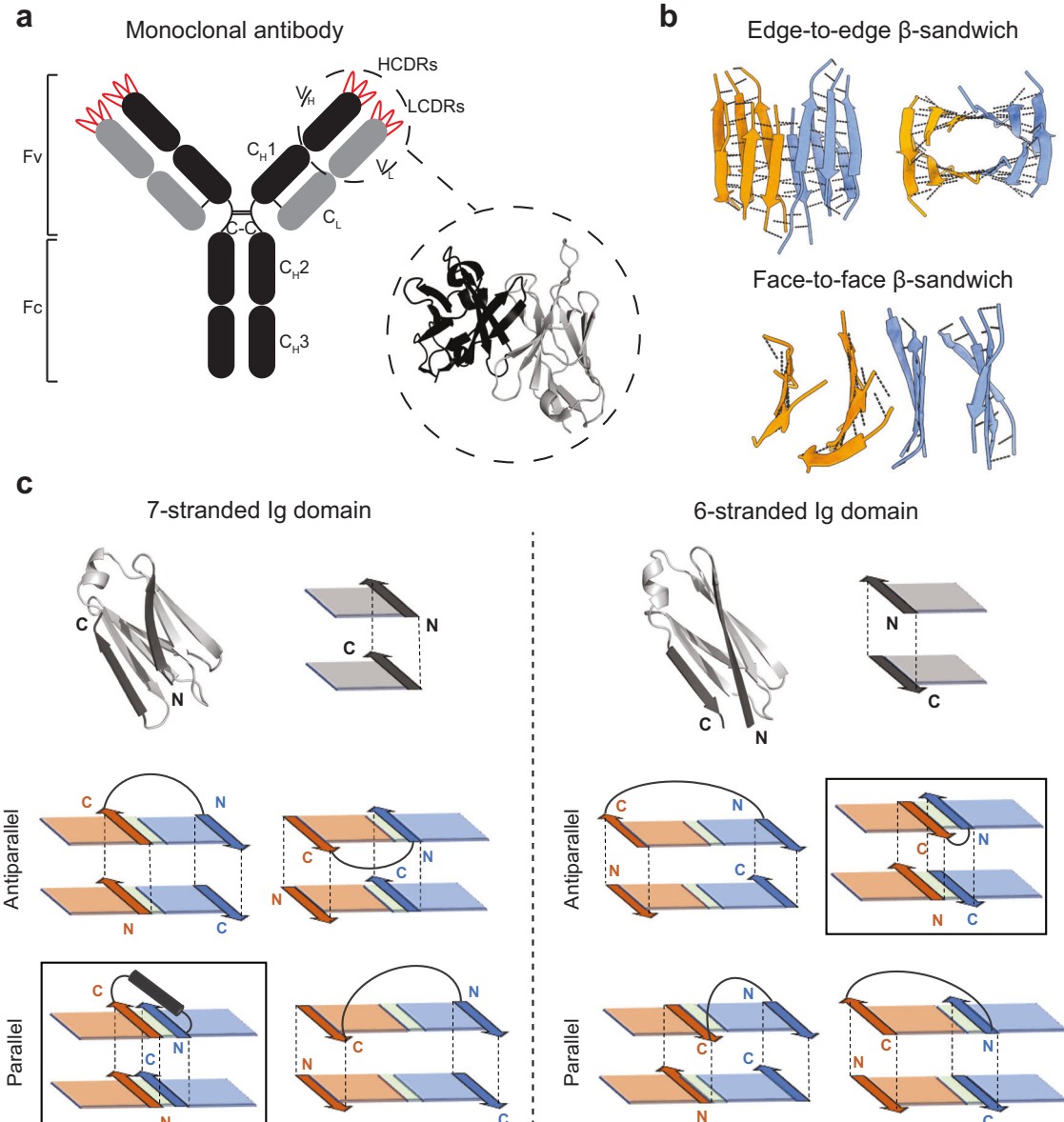

**Fig. 1 | Principles for designing edge-to-edge β-sandwiches from immunoglobulin domains. a** Schematic representation of the antibody molecule. Right inset shows the $V_H$-$V_L$ dimer as in scFvs (PDB 5YD3). **b** Cartoon representation of edge-to-edge and face-to-face β-sandwiches formed by two interacting immunoglobulin domains (orange and blue). Backbone hydrogen bonds between amine and carbonyl groups from adjacent β-strands are shown in dashed lines. In the face-to-face arrangement all edge β-strands are solvent exposed. **c** On the top, de novo designed immunoglobulins dIG8-CC (left) and dIG14 (right) with their corresponding sketch. On the bottom, schematic representations of all possible edge-to-edge arrangements having the C- and N-terminal β-strands of the first and second Ig domains, respectively, in the same layer as needed for a sequence-local fusion of a pair of 7-stranded (left) and 6-stranded (right) Igs. Interdomain hydrogen-bonded pairing is colored in green. Squared representations are the optimal fusion arrangements.

assembly based on blueprints[23] sampling helix lengths ranging between 12 and 16, in combination with loop lengths and geometries strongly favoring the resulting βαβ motif. Previous studies analyzing naturally occurring proteins identified the most common αβ and βα loop geometries, and how these are coupled with the lengths of secondary structure elements for correctly folding βαβ super-secondary structures[23,24]. In addition, to maximize hydrogen bond pairing between the N- and C-terminal β-strands not involved in the rigid fusion (i.e., the second edge-to-edge interface interaction) we considered different extensions of the N-terminal β-strand (from 0 to 3 residues). For the generated backbones, we performed flexible-backbone Rosetta sequence design[27,28] calculations on the inserted residues and surrounding ones located in β-strands. We mutated core cysteine residues, which formed a disulfide bond in each domain, to

ease the folding process and avoid dependence of protein stability on changes in redox conditions (an advantage over antibody-based formats). We found that the designed interdomain connections having more optimal interface interactions and lower backbone strain were those consisting of a 14-residue α-helix flanked by αβ and βα loops with specific ABEGO backbone torsions—i.e., BA/GB/AGB/GBA for the αβ loop, and BAB for the βα loop. Folding of the designs was then assessed by AlphaFold2 (AF2)[29,30] structure prediction, and identified 14 designs with highly confident predictions (pLDDT > 90) that matched very closely the designed structures (Cα-RMSD < 1 Å) (Supplementary Table 1). The designs were further validated by 9-mer fragment quality analysis on the designed region for assessing the local sequence-structure compatibility[19,23]. To further corroborate that the designed connections correctly guide folding towards the target

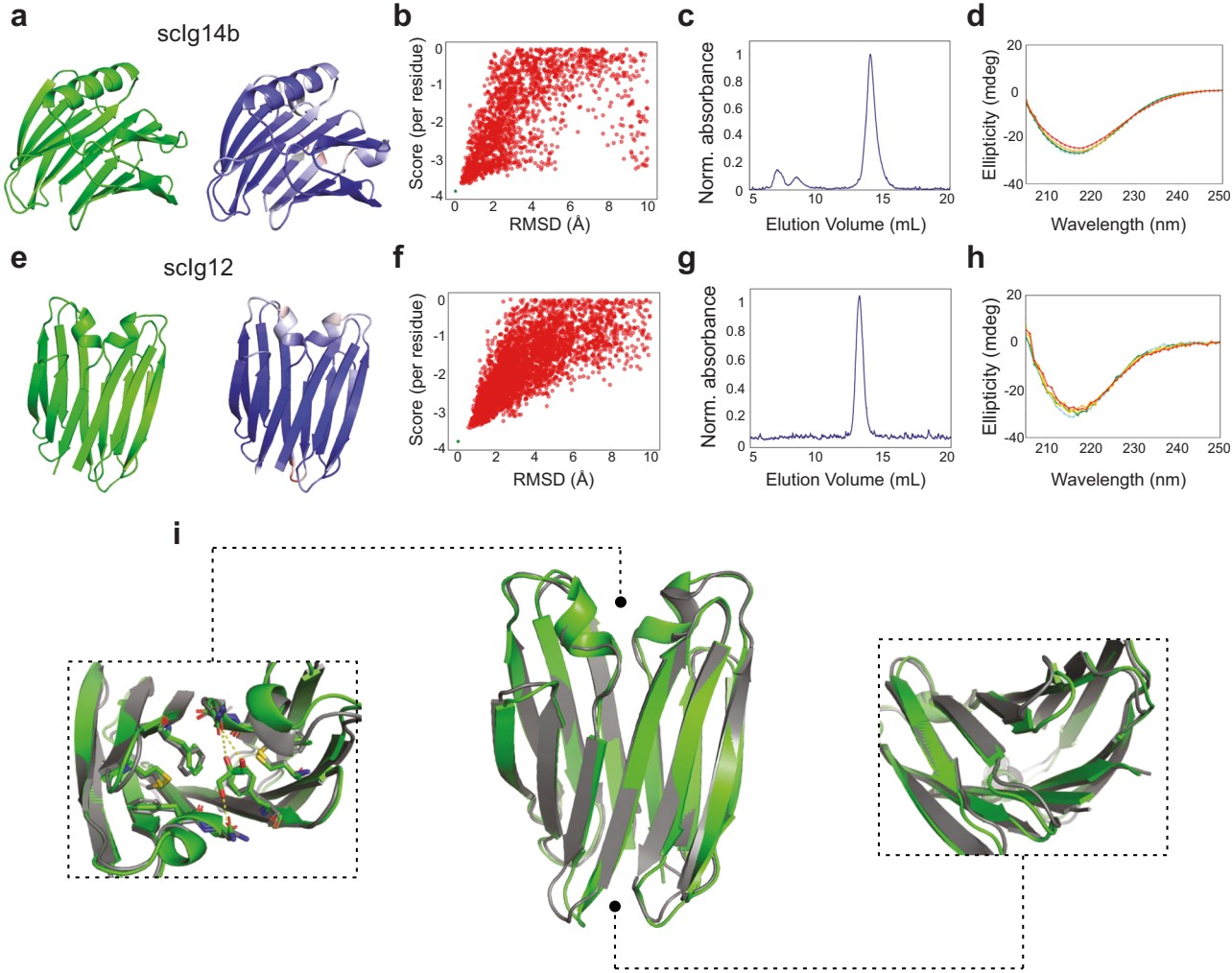

**Fig. 2 | Design and experimental characterization of edge-to-edge single-chain immunoglobulin dimers. a, e** scIg14b and scIg12 design models (in green, left) and their corresponding first AlphaFold2 model (right) colored by pLDDT; color scale goes from red (70) to blue (100). pLDDT values range from 72 to 98 and from 72 to 99 in (**a**) and (**e**), respectively. **b, f** Rosetta folding simulations for each of the designs. Y-axis indicates the per-residue Rosetta score and X-axis Cα-RMSDs against the design. Green stars correspond to the per-residue Rosetta score of the designs. **c, g** SEC-MALS chromatograms of purified designs. Estimated molecular weights for scIg14b and scIg12 confirm their monomeric state; see Supplementary Table 2. **d, h** Far-ultraviolet circular dichroism spectra at increasing temperatures (aqua: 25 °C, green: 40 °C, orange: 60 °C and red: 90 °C). **i** scIg12 design model (green) in comparison with the crystal structure (gray). Left and right insets show top and bottom views, respectively, of the structural agreement across the two β-sheets. A small ligand-binding cavity is formed at the top via hydrogen bond and aromatic interactions at the interdomain interface. Source data are provided as a Source Data file.

orientation, we performed Rosetta ab initio folding simulations[31] using intradomain constraints to focus sampling on the designed interdomain connections.

We selected the three designs with the most optimal sequence-structure agreement for experimental characterization (Fig. 2a, Supplementary Figs. 2, 3): scIg14a (BAB-H14-GBA), scIg14b (BAB-H14-GBA) and scIg14c (BAB-H14-BA). We ordered synthetic genes encoding for the designed sequences, expressed them in E. coli, and purified them by affinity and size-exclusion chromatography. All three designs were found to be well-expressed, soluble and monomeric by size-exclusion chromatography combined with multi-angle light-scattering (SEC-MALS) (Fig. 2c, Supplementary Figs. 3c, g, 4; Supplementary Table 2). Moreover, they were found to have far-UV circular dichroism spectra characteristic of their folds and turned out to be hyperstable by circular dichroism (Fig. 2d and Supplementary Fig. 3d, h): the proteins remain folded either at 90 °C or in 7 M GdnCl (Supplementary Fig. 5).

For the antiparallel design, we started from the dIG14 crystal structure (PDB 7SKP), where two de novo Ig monomers show antiparallel pairing between the N-terminal β-strand of one domain and the sixth β-strand of the other monomer. As the seventh (C-terminal) β-strand of each domain was found to be partly disordered and not present in the crystallographic dimer interface, we reasoned that the C-terminal region could be removed and then generated single-chain dimers of 6-stranded Ig monomers connected by a short β-hairpin loop; overall building a 12-stranded β-sandwich (Fig. 1c, right). To this aim, we removed the last nine residues of each domain, and designed a short loop connection (between 2 and 5 amino acids) between different anchoring residues (W68 or G69 of one monomer with G1 or R2 of the other monomer) by Rosetta fragment-based insertion. We found that the gap could be easily bridged with minimal backbone strain with loops of two or more residues connecting G69 with G1. For the designed fusions, AF2 generated highly confident predictions (pLDDT > 90) across all residue positions and that matched very closely the design model (Cα-RMSD of 0.6 Å) (Fig. 2e and Supplementary Table 3). The design with a two-residue hairpin loop (scIg12) was also found to have a strongly funneled energy landscape by Rosetta ab initio constrained folding simulations (Fig. 2f). We selected scIg12 for experimental characterization and, as the three scIg14 designs, was

found to be well-expressed in E. coli, soluble, and monomeric by SEC-MALS (Fig. 2g). It had a far-UV circular dichroism spectra characteristic of all-β proteins and was stable at 95 °C (Fig. 2h) or 7 M GdnCl (Supplementary Fig. 5). We succeeded in solving a crystal structure of scIg12 at 2.8 Å resolution by molecular replacement using the design model (Fig. 2i, Supplementary Table 4), and was found in excellent agreement with the computational model across the 12 β-strands (Cα-RMSD of 0.8 Å). A minor deviation was observed in the conformation of the designed connection. At the bottom, the designed linker is surrounded by a tightly packed area stabilized by aromatic stacking, and at the top it was found a cavity binding a glycerol molecule (as crystallization component) that is surrounded by the two β-arch helices (Fig. 2i, insets).

## Structural analogs of single-chain immunoglobulin dimers in nature

Although β-sandwiches are ubiquitous in nature, the complex β-sheet arrangement of two-layer single-chain Ig dimers leads to protein topologies— i.e., number of strands, their connections, and pairings (parallel or antiparallel)—that are hard to find in nature. We searched for structural analogs of the scIg12 and scIg14 designs in the PDB and the AlphaFold Protein Structure Database[32] (AlphaFold DB version Nov 2022) using the Foldseek server[33]. The top hits found in the PDB were β-sandwiches with a lower number of β-strands and different strand pairing organization—the top hit is a 9-stranded β-sandwich (PDB 4JOX) with a low TM-score[34] of 0.52 (Supplementary Fig. 6b). In the AlphaFold DB, the top-hit model retrieved by Foldseek was an 8-stranded β-sandwich of low confidence (pLDDT 67), also with a TM-score of 0.52 (Supplementary Fig. 6c). We manually inspected other hits from the AlphaFold DB and only found three models with topologies partially similar to our designs and connecting two Ig domains in alternative ways (Supplementary Fig. 7). First, a multidomain protein (Uniprot accession number K0ESF4) containing a single-chain Ig dimer (predicted with high confidence; pLDDT of 96), where both domains have edge-to-edge pairing only in one layer and are fused through a β-arch connection that rotates one domain with respect to the other (Supplementary Fig. 7a); having a TM-score of 0.51 with our designs. In the second hit (Uniprot accession number P81333), two interacting Ig domains are fused through an extended loop connection that prevents edge-to-edge pairing (Supplementary Fig. 7b). The third hit (Uniprot accession number A0A0S6VQQ4) is the closest analog to the scIg12 design (TM-score of 0.69), and corresponds to a two-layer single-chain Ig dimer (although predicted with low confidence; pLDDT of 69) that inserts a subdomain between two Ig domains, and has suboptimal edge-to-edge pairing due to a register shift in one of the two layers (Supplementary Fig. 7c).

## Predicted folding of edge-to-edge β-sandwiches is favored over face-to-face dimers

Having predicted that folding of edge-to-edge, single-chain Ig dimers can be strongly encoded by sequence, we next investigated whether the alternative face-to-face orientation, as seen in the variable region of antibodies, can be designed with similar predicted folding properties. Using two copies of the 7-stranded dIG8-CC domain, we first generated face-to-face dimer interfaces by sampling rotations and translations allowing packing between the first β-sheet (which contains the N-termini) of one domain and the second β-sheet (which contains the C-termini) of the other domain. We then performed loop closure for those dimer configurations with both termini in enough proximity to be linked by short loops (between 2 and 3 residues). In the face-to-face arrangement, the loop connecting both domains is a β-arch, as the two connected strands are in opposing β-sheets, and is, therefore, a non-local connection[17]. Closed structures were then subjected to sequence design calculations on the generated interfaces (Fig. 3a and Methods). Among the best interacting interfaces, we identified 19 designs with

excellent local sequence-structure compatibility, structured β-arches, and confident AF2 structure predictions (pLDDT > 90 and Cα-RMSD < 1.5 Å). For these, we performed Rosetta ab initio folding simulations using solely intradomain constraints (i.e., leaving unconstrained the interdomain contacts and the connecting loop) to assess the efficiency of the designed β-arch connections and interfaces in correctly folding the single-chain dimeric structure. As depicted in Fig. 3b (and Supplementary Fig. 8), although the designs are generally predicted as the lowest energy structure (in agreement with AF2), the energy landscapes are less funneled than in the edge-to-edge designs described above (Fig. 2b, f and Supplementary Figs. 3b, f, 9). Also, these folding simulations evince lower energy gaps between the designed conformation and competing ones (i.e., those with large Cα-RMSD and close in energy), which tend to have more open hydrophobic interfaces (Fig. 3b).

## Fusion of de novo immunoglobulin domains through functional binding loops

Having determined the hyperstability of edge-to-edge single-chain dimers, we next sought to assess their ability to scaffold functional loops at the interface between domains. We computationally grafted an EF-hand calcium-binding motif (DKDGDGYISAAE; PDB 1NKF[35]) in the β-hairpin bridge of scIg12 (loop connecting both domains) by designing N- and C-terminal linkers (short and structured) to integrate the motif into the scaffold (see Methods). We generated designs with loop insertions (i.e., EF-hand plus linkers) containing between 14 and 23 amino acids, and subsequently analyzed them with AF2 and Molecular Dynamics (MD) simulations. The designs are named as scIg12 + EFxy, where x refers to the β-hairpin number of the loop insertion (three in this particular case) and y to the loop sequence. We first probed protein folding by checking whether AF2 predictions closely matched the structure of the β-sandwich scaffold (i.e., not considering loop insertions). More than one-third of the grafted designs had at least one (out of five) high-confidence AF2 prediction (pLDDT > 90) with a Cα-RMSD < 1 Å to the scaffold, and for 22 designs all five predictions were within these ranges—convergence among the five models is another confidence metric that correlates with design success[36]. Among them, we identified four designs with AF2 predictions recapitulating the design structure across the scaffold and the inserted loop (0.7 Å < Cα-RMSD < 2.7 Å and 77.9 < pLDDT < 96.3 across the five models; Fig. 4a and Supplementary Fig. 10a). As expected, the AF2 structural deviations from the design are higher on the EF-hand motifs, which is likely associated with a higher flexibility. To get further insight into the structural preorganization of the four designed binding motifs, we carried out MD simulations (three replicas of 500 ns each) in the absence of calcium (Fig. 4b, c and Supplementary Fig. 10b, c). Although the binding motif region was generally found more flexible than the scaffold, for three designs the loop orientation and conformation remained close to the designed structure; suggesting high structural preorganization—a previous study scaffolding similar EF-hand motifs stressed the importance of the stability of such loops for ligand binding purposes[37]. The binding loop in the fourth design (scIg12 + EF3d) showed very high flexibility and severe deviations from the designed structure that correlate with the loss of a salt bridge stabilizing the joint between the EF-hand motif and the N-terminal linkers. All four designs incorporated a salt bridge between an aspartate or glutamate on the N-terminal motif linker and a lysine of the EF-hand motif two positions ahead; and for the three stable designs the pairwise distances between the salt-bridging residues remained within a close interaction distance throughout the simulations (Fig. 4d and Supplementary Fig. 10d).

Among the three most rigid designs identified by MD, we selected scIg12 + EF3a (loop sequence KDDKDGDGYISAAEK) for experimental characterization as it was the design with AlphaFold2 predictions of highest confidence (pLDDT > 90 for the five models and across all

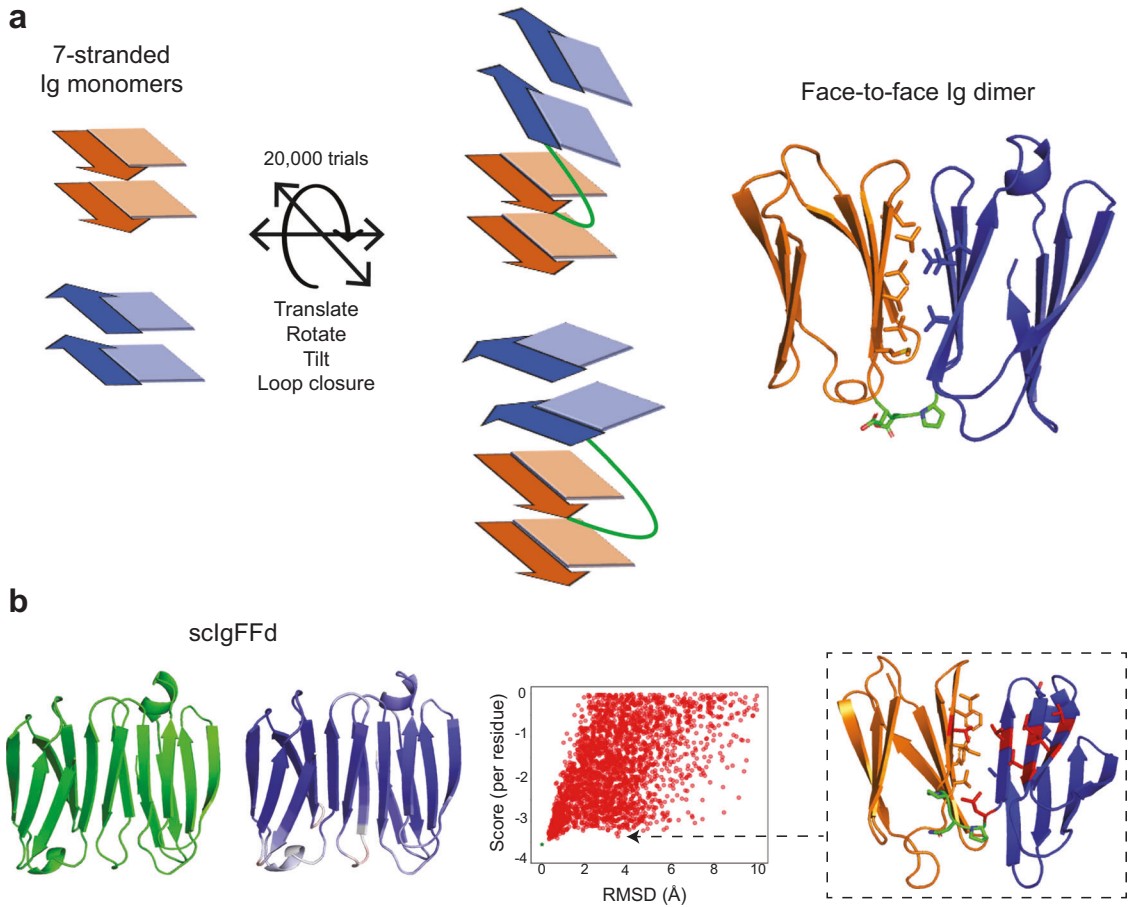

**Fig. 3 | Design and in silico validation of face-to-face single-chain immunoglobulin dimers. a** Single chain face-to-face Ig dimers were generated by fusing two 7-stranded domains. Multiple translations, rotations, and tilts were sampled before loop closure. N- and C-terminal domains are displayed in orange and blue, respectively, and the designed closing loop is in green. **b** From left to right, representative face-to-face single-chain dimer design model together with its AF2 prediction, colored by pLDDT (values range from 82 to 98); color scale goes from red (70) to blue (100). Constrained folding simulations indicate difficulties in achieving funnel-shaped energy landscapes. Green stars represent the Rosetta energy score of the designs. Right inset exemplifies alternative misfolded conformations close in energy to the design and exposing part of the hydrophobic interface to the solvent. Side chains are shown for residues in the linker and in the face-to-face interface. In red, solvent-exposed interfacial hydrophobic residues. Source data are provided as a Source Data file.

residue positions) and most similar to the design (Cα-RMSD < 1 Å) (Fig. 4a, Supplementary Table 5). We also noted by MD that the structural rigidity of the binding motif increased in the presence of calcium and that the interactions of the four EF-hand binding residues (D71, D75, and E82 through their acidic sidechain and Y77 through the backbone oxygen atom) remained nearly constant throughout the simulation (Fig. 4e, f). We ordered a synthetic gene encoding for the design, and was found to be well-expressed in E. coli, soluble, and monomeric by SEC-MALS (Fig. 4g). Moreover, it was found stable at 90 °C and at 7 M GdnCl, as the parent scIg12 design (Fig. 4h and Supplementary Fig. 5). We next carried out time-resolved terbium luminescence experiments to determine the metal binding ability of the design. Besides calcium, EF-hand motifs can also bind terbium, which has a similar ionic radius and coordination sphere. Terbium provides a convenient binding readout, as its fluorescence can be sensitized by energy transfer from proximal aromatic residues (e.g., tyrosine or tryptophan) when excited at 280 nm. The design was found to bind terbium in a concentration-dependent manner and with a characteristic terbium fluorescence spectrum with an intensity peak at 544 nm (Fig. 4i). Terbium-binding titrations were monitored at 544 nm with scIg12 + EF3a at 10 μM final concentration, and showed a hyperbolic increase in luminescence that was fitted to a 1:1 binding model, which gave an estimated $K_d$ of 5.1 μM (Fig. 4j). To probe calcium binding, we performed a competitive binding assay by titrating calcium and

monitoring the decrease in the terbium fluorescence (of scIg12 + EF3a mixed with terbium at 10 μM). Calcium was found to compete with terbium for the same binding site (Fig. 4k), but with much lower affinity ($K_d = 1.3$ mM)—this is expected given the lower charge density of calcium compared to terbium and hence weaker electrostatic interactions with the EF-hand motif. Lastly, we succeeded in solving the crystal structure of scIg12 + EF3a in a complex with terbium at 2.8 Å resolution by molecular replacement using the design model (Fig. 4l, Supplementary Table 4). The structure of the scaffold was found nearly identical to the design model (Cα-RMSD of 1.2 Å) and the parent scIg12 crystal structure (Cα-RMSD of 0.7 Å), which underscores the robustness of the extended β-sandwich scaffold to a 15-residue loop insertion (Supplementary Fig. 11). We identified electron density around the designed metal position that was unambiguously assigned to a Tb³⁺ ion (Fig. 4l, inset), as confirmed by anomalous X-ray scattering data (Supplementary Fig. 12), but the binding loop backbone could not be traced due to missing electron density; pointing to loop flexibility even with bound metal.

## Scaffolding multiple binding loop combinations in single-chain Ig dimers

Proteins able to scaffold multiple functional loops are particularly relevant for achieving increased or bispecific activities. Multiple loops located on the same side of the scaffold could together build large

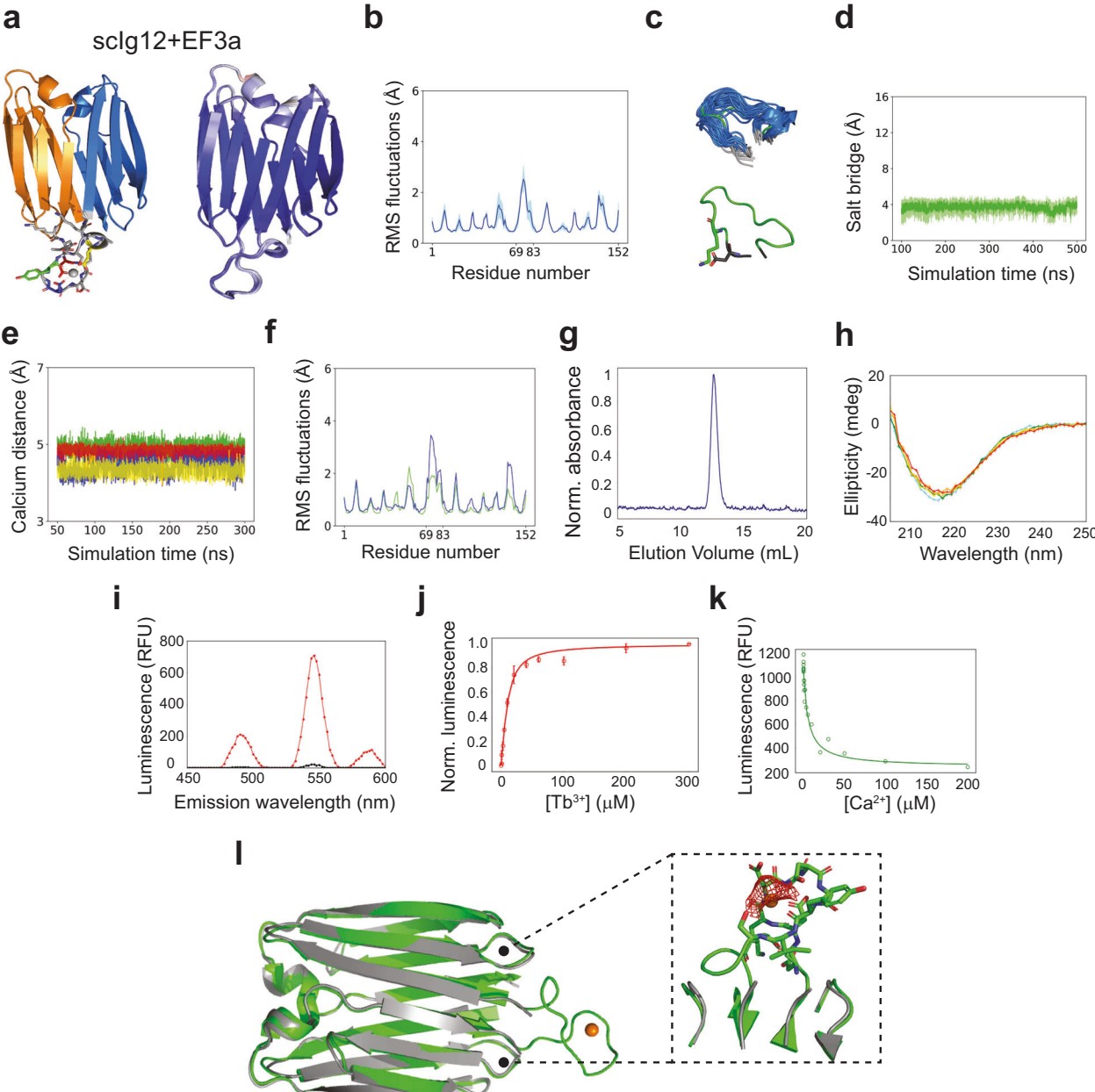

**Fig. 4 | Computational design and characterization of a metal-binding loop on scIg12. a** Left, computational model of scIg12 + EF3a with a grafted EF-hand motif (gray) and Ca²⁺ (gray sphere) interacting with the EF-hand motif chelating residues, which are D71 (red), D75 (blue), Y77 (green), E82 (yellow). N- and C-terminal Ig domains are colored in orange and blue, respectively. Right, AlphaFold2 predictions of scIg12 + EF3a. Residues are colored from red to blue by increasing pLDDT (scale 70–100). **b** Averaged root mean square fluctuations (RMSFs) obtained from molecular dynamics simulations (MD) in the absence of calcium (3 replicas of 500 ns). **c** Conformational ensemble of the grafted motif obtained from MD simulations (top) and representation of its anchoring lysine shown as sticks (bottom). **d** Pairwise distances between the salt bridge residues of the grafted motif monitored by MD. **e** Distances between EF-hand motif chelating residues and calcium monitored by MD. The four series match the colors of binding residues as in (**a**). **f** RMSFs obtained from a 300 ns MD in the presence (green) and absence (blue) of calcium. **g** SEC-MALS chromatogram of purified scIg12 + EF3a. The estimated

molecular weight confirms the monomeric state; see Supplementary Table 2. **h** Far-ultraviolet circular dichroism spectrum at increasing temperatures (aqua: 25 °C, green: 40 °C, orange: 60 °C, and red: 90 °C). **i** Tb³⁺ concentration-dependent time-resolved luminescence intensity of 10 μM scIg12 + EF3a using excitation wavelength $\lambda_{ex}$ = 280 nm and emission wavelength $\lambda_{em}$ = 544 nm. **j** Terbium binding titration at 10 μM final concentration. Normalized time-resolved luminescence intensities are displayed. The means ($n$ = 3) are fit to a one-site binding model by non-linear least squares regression ($K_d$ = 5.1 μM). Error bars present the standard deviation of the mean of the three replicates. **k** Time-resolved luminescence intensity in relative fluorescence units (RFU) for Ca²⁺ titrations with 10 μM scIg12 + EF3a and 10 μM Tb³⁺, showing Ca²⁺ competition with Tb³⁺ for the scIg12 + EF3a binding site. **l** scIg12 + EF3a design model (green) in comparison with the crystal structure (gray). Inset shows unambiguously assigned electron density around the designed metal position (orange sphere). Source data are provided as a Source Data file.

binding interfaces with targets, as complementarity-determining regions in antibodies or nanobodies. Functional loops displayed on opposite sides instead could simultaneously bind two different targets, which can trigger a wide range of mechanisms. To explore the

potential of single-chain Ig dimers to host multiple loops, we next assessed the ability of scIg12 to simultaneously scaffold two EF-hand motifs in multiple combinations of positions. We focus on β-hairpin positions since scIg12 + EF3a showed that the β-hairpin bridge between

both monomers is a sweet spot for functionalization, and β-hairpin positions are generally more favorable in terms of folding than β-arch loop regions. scIg12 has four β-hairpin positions on the top side (two from each monomer) and another one on the bottom side (bridging both monomers). To generate designs with two functional loops, we sought to design additional loop insertions individually at different β-hairpin positions, and then combined these in pairs. To this end, we started computationally grafting EF-hand motifs (by following the same approach described above) at the other two unique β-hairpin positions, on the top side of the scaffold: β-hairpins connecting strands 1–2 (EF1) and 3–4 (EF2) from the N-terminal domain—β-hairpins connecting strands 7–8 and 8–9 on the C-terminal domain are equivalent, respectively, due to the internal symmetry of the scIg12 scaffold (Fig. 5a). We generated nine designs with individual loop insertions (four and five designs for the first and second β-hairpin, respectively; Supplementary Fig. 13) having accurate and confident AF2 predictions (Cα-RMSD < 2.6 Å and pLDDT > 85, across all five predictions). The designed binding loops were diverse in terms of sequence, length (15–23 residues), and amino acid composition (Fig. 5b). We selected EF1a and EF2e, as the best designs for both positions for subsequent combination in pairs (Fig. 5a, Supplementary Fig. 13, Supplementary Tables 5, 6). Given the internal symmetry of scIg12, we ported EF1a to the fourth β-hairpin to obtain EF4 for subsequent combination in pairs. For a direct combination of individual designs, we verified that both loops were compatible, particularly for those on the same side.

For experimental characterization, we selected three designs displaying loops on opposite sides (scIg12 + EF1a + EF3a, scIg12 + EF2e + EF3a, and scIg12 + EF3a + EF4)—all of them sharing the EF3a motif on the bottom side and the second loop scanning three β-hairpin positions on the top side—and one design with two loops on the top side (scIg12 + EF1a + EF4) (Fig. 5c, d). The four designs were well-expressed in E. coli, monomeric by SEC-MALS (Fig. 5e and Supplementary Fig. 4) and thermostable by circular dichroism (Supplementary Fig. 14). The four designs bound terbium in a concentration-dependent manner with apparent affinity constants ($K_d$) in the 8–12 µM range (Fig. 5f), which are similar to that of scIg12 + EF3 with one binding loop. At saturating concentrations of terbium (200 µM), the luminescence intensities of the four designs were compared to that of scIg12 + EF3a at equimolar concentrations (5 µM) and showed a 1.7–2.3 fold increase (Fig. 5g). This luminescence enhancement confirmed that the two loop insertions simultaneously bind terbium across the four designs. Given the diversity of designed loop insertions and explored positions, the stability and binding properties of all tested designs underscores the robustness and versatile potential of scIg12 to simultaneously scaffold multiple functional loop combinations either oriented on opposite or same sides.

### Structural space of extended immunoglobulin β-sandwich scaffolds

Having structural control over such robust β-sandwich scaffolds can be particularly relevant when designing protein-binding interfaces engaging multiple loops. The topology and scaffold structure of single-chain Ig dimers control the number and relative position between loop sites that can be functionalized. For example, 12- and 14-stranded β-sandwich scaffolds (as the scIg12 and scIg14 designs) have one and two β-hairpin sites on the bottom, respectively, and four β-hairpin sites on the top, but with different relative distances. The design principles we have described above apply for generating single-chain dimers from (previously optimized) Ig domains undergoing minimal structural modification, and through short and rigid connections. However, higher diversity in terms of scaffold structure and spatial distribution of functionalization spots could be achieved through more irregular or larger connections, yet structured, interfaces. We sought to explore the structural space of extended Ig β-sandwich scaffolds with deep-learning hallucination using the AF2 network, as it efficiently generates structural diversity with minimal prior information[38], and analyzed how the geometry of Ig domains and their edge-to-edge interfaces can tune the number and positioning of hairpin sites. To this end, we combined in pairs a set of de novo Ig monomers, and hallucinated the four terminal strands (i.e., those involved in the two-layer, edge-to-edge interface) and a range of interdomain connections (variable in length), while keeping the sequence of the rest of positions fixed. We also considered insertions or deletions on the terminal strands to further enhance structural diversity. Hallucinated protein backbones with two-layer, edge-to-edge interfaces were then sequence-optimized with ProteinMPNN[39] on the hallucinated positions, and a total of 959 designs with high-confidence AF2 predictions (pLDDT > 90 and RMSD < 1 Å) were obtained for analysis.

The sequences are diverse both in terms of interdomain connection geometry and structure of the Ig domain components (Fig. 6a), and constitute extended β-sandwiches containing between 12 and 16 β-strands. We found that Ig domain pairs could be combined and properly fused through a wide variety of mechanisms contributing to scaffold diversification. First, the number of strands can be tuned by adding or removing interface residues. For example, 12- and 13-stranded scaffolds were generated from 7-stranded Ig domains by removing one terminal strand in each layer or in any of the two, respectively; and connecting them with a short hairpin loop. On the contrary, 15- and 16-stranded scaffolds added one or two (one in each layer) strands in the interdomain connection through extra β-hairpin loops. Importantly, extending β-sandwiches from 12 to 16 strands gradually separates β-hairpin sites from different domains, and increases the number of available β-hairpins in the interface area that could cooperate in multivalent binding (Fig. 6b). Second, extending terminal strands in the linker region can change the register of strand pairing at the interface between domains, controlling the translation (along the strand direction) of one domain relative to the other. Third, changes in the curvature of edge strands can shorten the distance between the N- and C-termini of Ig domain pairs that otherwise would be incompatible for rigid fusion. Overall, the structure of extended β-sandwiches can be highly diversified at the level of Ig domain components, interdomain connections, and both number and positioning of potential functional sites.

## Discussion

Traditional protein engineering strategies have successfully repurposed naturally occurring proteins to perform new functions of interest, but they are often limited by existing structures that are hard to modify or adapt without a loss of stability, solubility, or any other property. Antibody-based formats harnessing the binding properties of natural variable regions have a long-standing dependency on the structure of the natural Fv Ig framework that limits their biophysical properties. Here, we showed that Ig frameworks can be designed through two-layer edge-to-edge dimer interfaces in multiple orientations compatible with a stable fusion. The designed single-chain Ig dimers were found to be hyperstable scaffolds (they remain folded at 95 °C or 7 M GdnCl) with the robustness for displaying multiple ligand-binding loops, both individually and in combination, across all β-hairpin sites. Although we targeted the functionalization on the β-hairpin sites, due to folding reasons, the robustness of the scaffold should also enable functionalization on β-arches. The robustness of the scIg12 scaffold was borne out by nearly identical crystal structures obtained in the absence and presence of a 15-residue binding-loop insertion (Supplementary Fig. 11). Key to such robustness is the two-layer edge-to-edge strand pairing at the interface between Ig domains, which involves a high number of backbone-backbone hydrogen bonds and protects oligomerization-prone β-sheet edges from solvent; overall favoring monomeric stability and solubility. This is particularly advantageous over the face-to-face arrangement, for which all edge strands are solvent-exposed by definition. Interestingly, the computed

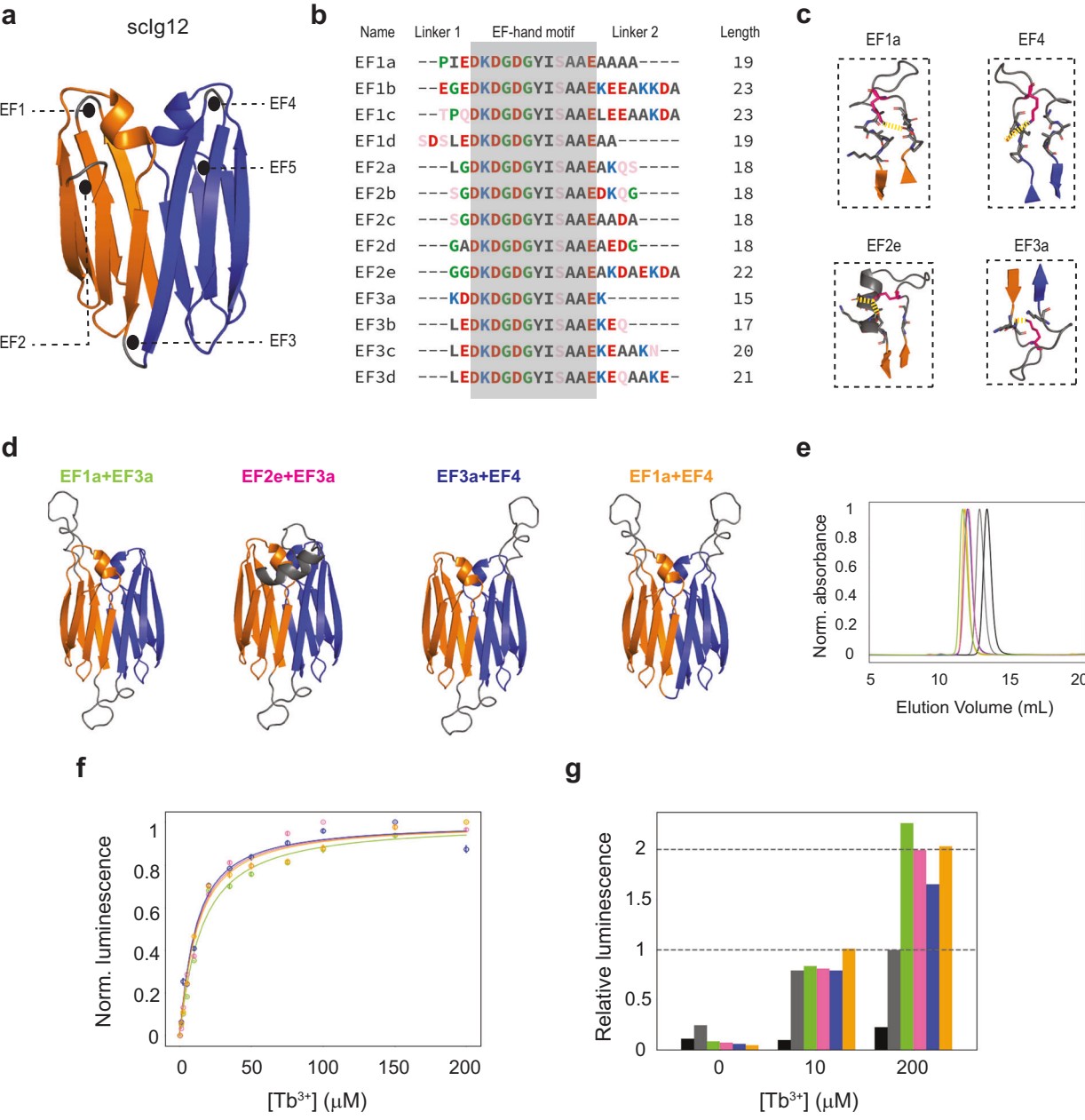

**Fig. 5 | Computational design and characterization of multiple binding loops on scIg12. a** The scIg12 scaffold has five β-hairpin regions, named as EFx, where x indicates their position in the sequence. While β-hairpins EF1 and EF2 come from the N-terminal monomer (orange), EF4 and EF5 derive from the C-terminal one (blue). EF3 corresponds to the linker bridging both monomers. Note that due to the internal symmetry of the scaffold, EF1 is equivalent to EF4, likewise EF2 to EF5. **b** Multiple sequence alignment of designed terbium binding loops (EF-hand motif and linkers) in silico validated. Sequences are colored according to amino acid types as follows: hydrophobic (dark gray), positively charged (blue), negatively charged (red), polar (pink), and turn-forming (green). **c** Closer view to the inserted loops. Salt bridges between K2 (from the EF-hand motif) and either D or E from the N-terminal linker are shown as yellow dashed lines. **d** The four experimentally characterized designs displaying two functional loops: EF1a + EF3a (green), EF2e + EF3a (pink), EF3a + EF4 (blue), and EF1a + EF4 (orange). **e** SEC-MALS chromatograms of the four designs (colored as in (**d**)) in comparison to scIg12 + EF3a (gray)

and their parental scIg12 (black). **f** Terbium binding titrations for the four designs (colored as in (**d**)) at 5 μM final concentration. Normalized time-resolved luminescence intensities are displayed. The means (n = 4) are fit to a one-site binding model by non-linear least squares regression, which gave as apparent $K_d$ of 12.3, 9.2, 8.3, and 9.6 μM for EF1a + EF3a (green), EF2e + EF3a (pink), EF3a + EF4 (blue) and EF1a + EF4 (orange), respectively. Error bars present the standard deviation of the mean of the four replicates. **g** Relative luminescence at different $Tb^{3+}$ concentrations with respect to the absolute luminescence of scIg12 + EF3a (dark gray) at a saturating concentration of $Tb^{3+}$ (200 μM). In black, relative luminescence of the non-functional scIg12 scaffold as baseline. The relative values were obtained as the ratio between two means obtained from the four replicates. All four designs with two functional loops (**d**) showed an approximated twofold increase over scIg12 + EF3a with a single functional loop at equimolar concentration (5 μM). Source data are provided as a Source Data file.

energy landscapes for edge-to-edge designs (scIg) were also more favorable than for face-to-face (scIgFF) designs, which in turn indicates challenges in preferentially stabilizing the latter. Such stability is remarkable given that all-β proteins have been historically more

challenging to design due to their aggregation propensity, and the scaffolds designed here are the largest monomeric all-β proteins designed to date. These scaffolds also represent protein topologies that are hard to find in nature, as the closest structural analogs

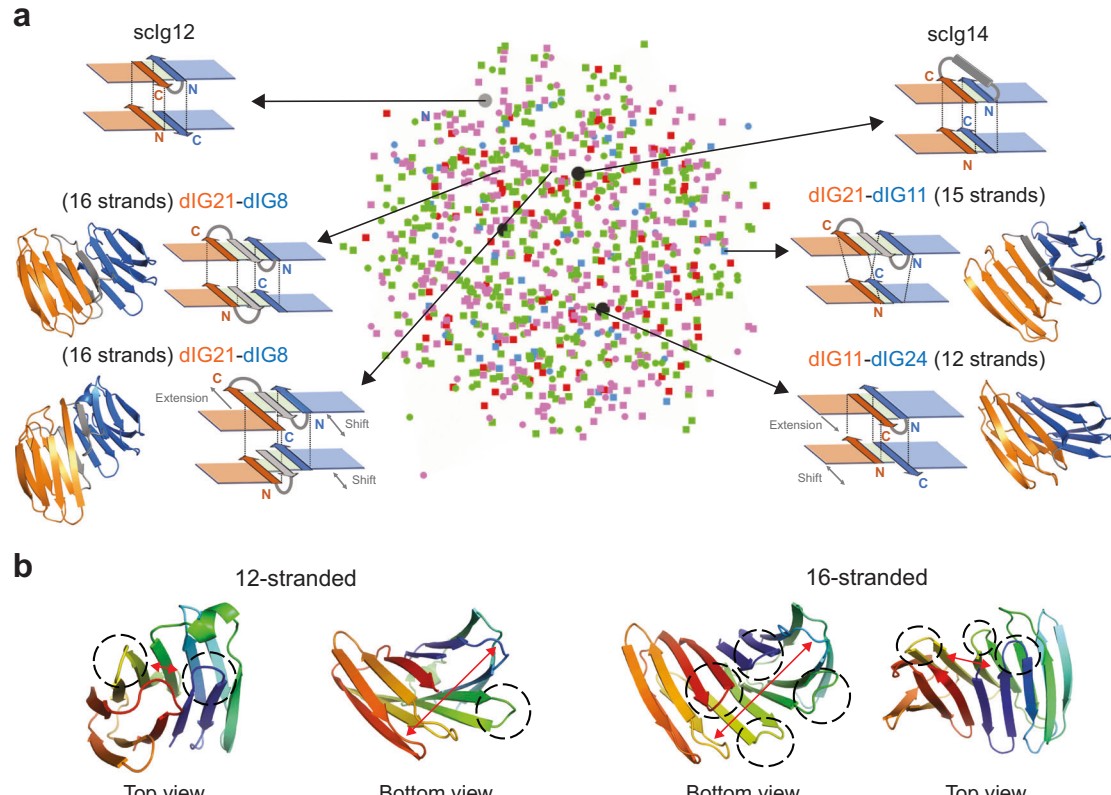

**Fig. 6 | Diversification of single-chain immunoglobulin dimers by deep learning hallucination. a** Structural similarity network of the high-confident 959 diversified scaffolds as a function of their TM-scores. Each node represents one design colored by the number of β-strands (12: green, 13: red, 15: cyan, and 16: pink). Larger black and gray dots indicate our scIg14 and scIg12 designs, respectively. Square nodes correspond to heterodimers, while circles to homodimers. The schematic representations show examples of diversified scaffolds. **b** The extension from 12 to 16 β-strands increases the number of functionalization spots, while gradually separates β-hairpin sites from different domains. The network was built and edited using Cytoscape[40].

correspond to proteins from the AlphaFold DB either having low-confidence predicted models, different interdomain organization or both.

From the molecular modeling perspective, the designed single-chain Ig dimer scaffolds fall in a size range (140–160 amino acids) that is quite challenging for ab initio structure prediction (without constraints guiding conformational sampling)—the gold-standard in silico test for de novo designed proteins during the last decade[19]. Remarkably, AF2 predicted high-confidence and accurate models for both edge-to-edge and face-to-face, non-functional designs without the need of multiple sequence alignment (MSA) information, which suggests that the designed sequences strongly encode for their structures. In contrast, the predictions for designs with ligand-binding loops were less confident on the inserted regions, especially those displaying two loops simultaneously. Yet, by combining protein structure predictions and MD simulations assessing ligand-binding loop structural rigidity, we accurately identified a series of functional designs having structured linkers—e.g., through salt bridge interactions— varying in length and sequence composition that enabled the correct integration of the EF-hand binding motif into our scaffolds. These functional designs turned out to be hyperstable in solution, while showing high-affinity terbium binding across multiple β-hairpin sites. They also showed substantially higher binding affinity (in the low μM range) than a EF-hand terbium loop we recently incorporated in a de novo Ig domain[18], likely due to improved predictions both in folding and structural rigidity of the designed loops (Supplementary Fig. 15; Supplementary Tables 5–7). Overall, the advent of deep-learning structure prediction breaks a historical size limit set by ab initio structure prediction and now allows us to make accurate structural validations for larger proteins than before. As shown here, these methods in combination with physics-based simulation techniques are expected to ease the accurate de novo design of increasingly large and complex folds, including multidomain proteins.

The robustness of the edge-to-edge single-chain Ig dimers for scaffolding multiple loops opens exciting possibilities in several directions. The designed binding loops (experimentally tested and/or in silico validated) were diverse in terms of sequence, length, amino acid composition, and scaffold position. Across three independent β-hairpin sites, the 12-residue EF-hand motif was correctly integrated in loops varying in length (ranging between 15 and 23 residues), and combining amino acids of all types—i.e., polar (S, T, Q, N), charged (K, E, D), aliphatic (A, I, L), aromatic (Y), and turn-forming (G, P)—as shown in Fig. 5b. Such length range is comparable, or larger, than the average length typically found in CDRs of antibodies[41,42]. Also, amino acid diversity in loop sequences is particularly relevant for designing protein-protein interactions[42], which require combining polar and hydrophobic interactions, and amino acids controlling loop conformational stability. The designed loops were structurally connected to the rest of the scaffold to a varying degree (Supplementary Fig. 13), which also influences the shape of epitopes that can be targeted. (e.g., protruding loops are in general better suited for targeting concave epitopes, as seen in nanobodies inhibiting enzyme activities[4]). Our results indicate that single-chain Ig dimeric scaffolds have the robustness to accommodate a wide range of loops in multiple sites (both individually and in combination), which holds promise for designing protein binding paratopes in a similar spirit to antibodies or nanobodies. In this regard, this work represents a stepping stone towards de novo designing antibody-like protein binding scaffolds.

Based on our encouraging results, it will be interesting in future work to design binding loops for protein targets of interest through high-throughput screening of loop libraries[43,44] (naive or computationally focused) or computational design. Indeed, recent advances in deep learning should facilitate the design of entirely new binding loops[45], without the need of grafting existing motifs and solely based on the target epitope structure. Furthermore, the tunability of the scaffold structure will be advantageous to provide loop conformational stability, and make additional contacts with protein targets (as in nanobodies[42] or monobodies[3]).

Single-chain Ig dimers may be suitable for diverse applications beyond binding single protein targets. The robustness of these scaffolds to simultaneously display loops in the two distal sides could be useful for creating compact, stable, and single-chain bispecific formats, which are particularly useful for co-localizing proteins of interest in a broad range of applications. Beyond loop scaffolding, single-chain Ig dimers could also shape binding pockets for small molecules or metal ions on the top (or bottom) of the interface between domains, where the structure of the scaffold and surrounding loops could be readily diversified to form cavities matching binding site geometries (the crystal structures of scIg12 and scIg12 + EF3a reveal a crystallization component binding to the top of the interface that is surrounded by several loops). The high stability of these scaffolds should ease the incorporation of cavity-creating mutations, as described for other de novo proteins[46]. Also, their extended β-sheet structures show potential as traps for extended peptides, as recently shown with other protein scaffolds recognizing amyloidogenic peptide segments through extended β-sheets leaving unpaired strands in the middle[47]. The versatile potential of single-chain Ig dimers opens exciting avenues worth exploring in future work.

## Methods

### Design of helical linkers for parallel single-chain Ig dimers

Chains A/D from the asymmetric unit of the Protein Data Bank (PDB) accession code 7SKO[18] were used as starting building blocks. We designed a 14 residue α-helix, flanked by short and structured βα and αβ linkers to connect both subunits. For the linkers, the most common loop geometries for βα and αβ connections (BAB-BA, BAB-GB, BAB-AGB, and BAB-GBA for βα-αβ[24] connections) were sampled. For each combination, 500 models were generated and those with undefined ABEGO types for either linker were discarded. This yielded a total of 1189 models, for which ten different sequences were generated with the FastDesign mover[28] implemented in RosettaScripts[48]. The resulting 11,189 designs were ranked according to their combination of: per residue Rosetta score, secondary structure shape complementarity score (between the α-helix and the packing 6th–9th β-strands), α-helix buried surface area and count of alanines in the α-helix. The top 2000 designs were selected for sequence-structure compatibility assessment and 308 of them picked for protein structure predictions with AlphaFold2. Out of the predictions, 14 designs showed excellent structural agreement and three of them were selected for experimental testing (scIg14a, scIg14b, and scIg14c) after removal of the disulfide bridges. For two of them, scIg14a and scIg14c, the N-terminal β-strand was extended by three residues in order to increase hydrogen bond pairing with the C-terminal β-strand.

### Design of linkers for antiparallel single-chain Ig dimers

We trimmed the seventh strand of the dIG14 structure (PDB code: 7SKP) and bridged the two domains by designing short loop segments using the Blueprint Builder[23] mover implemented in RosettaScripts[48]. Poly-glycine loop fragments ranging between two and five residues were inserted between W68 or G69 of one domain and R1 or V2 of the other domain. Fragment insertion was performed using the fldsgn_cen centroid scoring function.

### Design of face-to-face single-chain Ig dimers

The PDB accession code 7SKO[18] was retrieved and two copies of chain A were used as starting building blocks. First, both monomers were translated apart by a distance of 20 Å from their center-of-masses. Then, based on preliminary tests, a random rotation (100°–160°), translation (−15 Å to −5 Å) and tilt (−50° to −25°) was applied to one of the monomers prior fragment insertion of a 2–3 residue long linker in order to connect both domains. This protocol was repeated for 20,000 trials, generating 18,852 closed designs. As before, fragment insertions were performed using the Blueprint Builder[23] and the fldsgn_cen centroid scoring function implemented in RosettaScripts[48]. For each of these, the analyses were carried out in two subsequent rounds of sequence design, protein structure prediction, and fragment quality assessment as follows.

In the first round, 5 different sequences of the loop and the interface between domains as defined by the InterfaceByVector residue selector (cb_dist_cut = "11.0" and nearby_atom_cut = "5.5") were designed by the FastDesign[28] mover implemented in RosettaScripts;[48] yielding a total 94,260 unique models. The top 10,000 models based on their per residue score and interfacial buried surface area were selected for protein structure prediction. Out of the predictions, 370 models were selected for sequence-structure compatibility evaluation.

In the second round, 31 designs falling into the fragment quality filters underwent interface sequence optimization using ProteinMPNN[39], and the resulting unique sequences were passed for protein structure prediction. Out of the 1488 predictions for the optimized sequences, 74 designs were further selected, again, for sequence-structure compatibility evaluation, and 13 of them, alongside their parental designs (6), were picked for constrained folding simulations (total of 19).

### Design of functional loops

The PDB accession code 1NKF[35] was retrieved and only the 12 metal-binding residues (DKDGDGYISAAE) were kept. Blueprint files were generated for both the dIG14 single-chain dimer and the EF-hand motif. Additionally, 18 blueprint files were created for domain insertion by sampling all combinations of N-terminal linkers between 2 and 3 residues and C-terminal linkers spanning 0–8 residues. The N-terminal linkers were forced to be in an extended conformation, while the C-terminal linkers in α-helical conformation. RosettaRemodel[49] was run 100 times for each of the domain insertion blueprints while designing the sequence of both linkers; restricted to a series of residues specified in their corresponding blueprint files. The metal non-coordinating residues were allowed to repack while the metal-binding residues were kept fixed. 598 out the 1800 simulations produced at least one closed structure; resulting in a total of 772 unique EF-hand grafted single-chain models.

### Sequence-structure compatibility assessment

The local compatibility between the designed sequences and structures was evaluated based on fragment quality. For the design of linkers for parallel single-chain Ig dimers, sequence-structure pairs were considered locally compatible if for all residue positions at least one of the picked 9-mer fragments (based on sequence and secondary structure similarity with the design) had a RMSD below 1.0 Å. For the design of face-to-face single-chain Ig dimers, sequence-structure pairs were considered locally compatible if for all residue positions at least one of the picked 9-mer fragments (based on sequence and secondary structure similarity with the design) had a RMSD below 1.5 Å in the first round, and below 1.3 Å in the second round.

### Protein sequence design

All sequences designed by the FastDesign[28] mover implemented in RosettaScripts[48] were using the ref2015 scoring function. For the

interface sequence optimization ProteinMPNN[39] was used. Only interface residues as defined by the InterfaceByVector residue selector (cb_dist_cut = "11.0" and nearby_atom_cut = "5.5") were allowed to design with --num_seq_per_target 50, --sampling_temp "0.1", --seed 37 and --batch_size 1 flags activated, while fixing the rest of the sequence. All sequences were clustered at 100% identity with mmseqs2[50].

## Protein structure prediction

The local installation (LocalColabFold: github.com/YoshitakaMo/localcolabfold) of the optimized AlphaFold2 software version (ColabFold)[30] was used for protein structure predictions. For all protein structure predictions, the -use_turbo option was enabled for optimal speed, the number of recycles was kept at three (default), and the predictions were relaxed using the Amber force field. See following subsections for specific considerations.

Parallel and antiparallel single-chain Ig dimers: the predictions were made in the absence of MSA. For analysis purposes, Root Mean Square Deviations (RMSDs) of the scaffold (RMSD-total), RMSD of the βα loop (RMSD-βα), RMSD of the connecting α-helix (RMSD-helix) and RMSD of the αβ loop (RMSD-αβ) were calculated. Those designs whose predicted structures had RMSD value > 1.0 Å or pLDDT < 90 were discarded, yielding a total of 14 designs.

Face-to-face single-chain Ig dimers: for both rounds of design, protein structure predictions were made in the absence of MSA. For selection purposes, for each design a composite score = pTM * pLDDT * TM-score from the top3 AF2 predictions was computed. In the first round, all designs with a composite score ≤0.6 were discarded, yielding a total of 370 designs. In the second round, all designs with a composite score >0.7 were selected, yielding a total of 74 designs.

Functional loops: the predictions were made using MSAs as computed by the mmseqs2 method[50]. Several structural metrics from the predictions including the total Root Mean Square Deviation (RMSD-total), RMSD of the scaffold (RMSD-scaffold; excluding the inserted domain), RMSD of each of the linkers (RMSD-linker1/2) and the RMSD of EF-hand binding motif (RMSD-motif) against the grafted model were calculated for filtering purposes. First, all predictions with RMSDs > 1 Å (excluding RMSD-total) were discarded. This filter returned 351 predictions out of 3860 (772 grafted models * 5 AF2 predictions). Next, in order to assure prediction convergence, only those grafted models with at least four AF2 predictions passing these cutoffs were kept; yielding eight models in total. Finally, the best four designs according to their RMSD-total and pLDDT values were selected for further analysis.

## Molecular dynamics simulations

The four selected designs were used as starting points for MD simulations. Dodecahedron TIP3P water boxes were employed to solvate the proteins with a buffer distance of 11 Å to the box edges. Na$^+$ and Cl$^-$ ions were added to provide charge neutrality at a total concentration of 150 mM. The Amber14SB force field[51,52] was used for proteins and NaCl, and Ca$^{2+}$ parameters were obtained from ref. 53. First, the systems were minimized using the steepest descent method. Next, the systems were equilibrated involving an initial heating to 100 K at constant volume for 50 ps followed by heating to 298 K at a constant pressure of 1 bar. Production runs were performed with GROMACS[54] version 2018.3 at 1 bar and 298 K with periodic boundary conditions and a 2 fs timestep, with non-bonded short-range interactions calculated within a cutoff of 10 Å. Each of the four designs was simulated in three independent 500 ns replicates. For the simulations including Ca$^{2+}$, each of the two system conditions (absence/presence of Ca$^{2+}$) were simulated in one 300 ns run. The first 100 ns and 50 ns (absence/presence of Ca$^{2+}$) of each production run were discarded from analysis to allow for adequate equilibration from the starting conformation.

## Constrained folding simulations

All folding simulations were performed using the AbinitioRelax application implemented in Rosetta[26]. AtomPair constraints were defined for every intradomain Cα atom within a 8 Å distance cutoff and sampling was kept at 5000. The remaining options were as follows: -kill_hairpins, -use_filters true, -abinitio::rsd_wt_loop 0.5, -abinitio::rsd_wt_helix 0.5, -abinitio::rg_reweight 0.5, -abinitio:relax, -relax:fast, -abinitio::increase_cycles 10, -constraints:cst_weight 1.0. For plotting purposes, the contribution of the constraints term to the score was subtracted. Note that no constraints were used between edge-to-edge nor face-to-face interfaces.

## Deep network hallucination

We adapted the design approach described in ref. 38 for hallucinating edge-to-edge single-chain dimers from de novo Ig domains. We selected six de novo 7-stranded Ig monomers (dIG6, dIG11, dIG14, dIG21, dIG24) with high-confidence AF2 predictions, which were designed in our previous study[18], as a basis set for hallucinating single-chain dimers. For all pair combinations of these monomers, we hallucinated their N- and C-terminal β-strands (i.e., those involved in the interface) and a linker containing between 5 and 20 residues joining the two monomeric sequences, while keeping the rest of the amino acid sequence unchanged. Linker length ranges were selected to enable short connections through a β-hairpin loop or longer ones involving insertion of strands or α-helices. We also considered combinations involving 6-stranded versions of the monomers by removing the C-terminal strand. We ran five Monte Carlo hallucination trajectories (2000 steps) per design combination and used a loss function based on structure prediction confidence metrics (pLDDT and pTM) obtained from the AlphaFold2 network (model 4). Among all hallucinated sequence positions, those not present in the original monomeric sequences (i.e., inserted residues) were mutated at a mutation rate two times higher than the rest. All hallucinated models were filtered based on the geometry of their interfaces, and those forming two-layer edge-to-edge strand pairing were subjected to ProteinMPNN[39] sequence calculations on the hallucinated positions. We ran 20 ProteinMPNN[39] calculations per design and the resulting sequences were clustered with mmseqs2[50] using a 95% sequence identity cutoff.

## Recombinant expression and purification of the designed proteins

Synthetic genes encoding for the designed amino acid sequences were obtained from Genscript and cloned into the pET-28a-TEV expression vector, with genes inserted within NdeI and XhoI restriction sites and the pET-28a-TEV backbone encoding an N-terminal hexa histidine tag followed by a Tobacco-Etch-Virus peptidase (TEV) cleavage site. Escherichia coli BL21 (DE3) competent cells (Sigma) were transformed with these plasmids, and starter 10 ml cultures from single colonies were grown overnight at 37 °C in Luria-Bertani (LB) medium supplemented with 50 µg/ml kanamycin. Overnight cultures were used to inoculate 800 ml of LB medium supplemented with 50 µg/ml kanamycin and cells were grown at 37 °C under shaking, until an optical density (OD600) of 0.5–0.7 was reached. Protein expression was induced with 1 mM of isopropyl β-D-thiogalactopyranoside (IPTG) and cultures were incubated overnight at 18 °C. Cells were harvested by centrifugation 30 min at 5000 g and resuspended in 10 ml of buffer containing 30 mM Tris HCl, 250 mM sodium chloride, pH 8, 30 mM imidazole, and EDTA-free cOmplete Protease Inhibitor Cocktail (Roche Life Sciences). Cells were lysed using a sonicator and soluble protein was clarified by centrifugation 25 min at 25,000 g. Recombinant proteins were captured on nickel-sepharose HisTrap HP columns (Cytiva), which had been previously equilibrated with the same buffer. Recombinant protein purifications were performed with a gradient of 30-to-500 mM imidazole in the same buffer. Proteins were further purified by size-exclusion chromatography using a Superdex 75 10/300 GL (Cytiva)

column. The expression of purified proteins was assessed by SDS-polyacrylamide gel; and protein concentrations were determined from the absorbance at 280 nm measured on a NanoDrop spectrophotometer (ThermoScientific) with extinction coefficients predicted from the amino acid sequences using the ProtParam tool. Overall, we have obtained between 2 and 6 mg of purified protein per liter of cell culture without optimization of expression conditions.

## Circular dichroism
Far-ultraviolet circular dichroism measurements were performed to assess protein stability using a JASCO J-815 spectrometer at CCiTUB. Wavelength scans were measured from 260 to 200 nm using a 1 mm path-length cuvette. Protein samples were prepared in PBS buffer (pH 7.4) at a concentration of 0.3 mg/mL. Guanidine hydrochloride titrations were performed manually using a 7 M GdnCl stock solution dissolved into PBS.

## SEC-MALS
Size-exclusion chromatography coupled to multiple-angle light scattering (SEC-MALS) was performed in a Dawn Helios II apparatus (Wyatt Technologies) coupled to a SEC Superdex 75 Increase 10/3000 column (Cytiva). The column was equilibrated with 30 mM Tris·HCl, 250 mM sodium chloride, pH 8 at 25 °C and operated at a flow rate of 0.5 mL/min. Samples were simultaneously monitored with an UV detector (SPD-20, Shimadzu), a differential refractometer (OPTI-rEx, Wyatt Corp.), and a static multiangle laser light scattering detector (DAWN-HELEOS, Wyatt Corp.). Data processing and analysis proceeded with Astra 7 software (Wyatt Technologies).

## Protein crystallization and structure determination
Crystallization screenings were performed at the joint IRB/IBMB Automated Crystallography Platform. Screening solutions were prepared and dispensed into the reservoir wells of 96 × 2-well MRC crystallization plates (Innovadyne Technologies) by a Freedom EVO robot (Tecan) using the sitting-drop vapor diffusion method. scIg12 crystals appeared at 20 °C in drops consisting of 100 nL protein solution (at 15 mg/ml in 30 mM Tris pH 8, 250 mM NaCl) and 100 nL reservoir solution (45% v/v 2-methyl-2,4-pentanediol (MPD), 0.2 M calcium chloride, 0.1 M Bis-Tris at pH 6.5). scIg12 + EF3a crystals were obtained at 20 °C in drops consisting of 100 nL protein solution (at 20 mg/ml in 30 mM Tris pH 8, 250 mM NaCl) and 100 nL reservoir solution (20 % w/v PEG 2000, 0.1 M TRIS pH 7.0). Once the crystals appeared, 100 nL of crystallization solution enriched with 18 mM TbCl₃ was added to the drops. Both protein crystals were cryoprotected with a multi-component cryoprotectant (12,5% v/v diethylene glycol, 12.5% v/v MPD, 37,5% v/v 1,2- propanediol and 12,5% v/v dimethyl sulfoxide), harvested with 0.1–0.2 LithoLoops (Molecular dimensions), and flash-vitrified in liquid nitrogen.

X-ray diffraction data were collected at 100 K at the XALOC beamline of the ALBA synchrotron (Cerdanyola, Catalonia, Spain). For the scIg12 crystal, data were collected at wavelength 0.97926 Å, while for scIg12 + EF3a anomalous data were collected at wavelength 1.50102 Å and the difference anomalous map was calculated. The diffraction images were integrated using XDS[55] and scaled using Aimless[56]. The structures were solved by molecular replacement with MolRep[57] using the original design models. The structures were refined using Refmac5[58] and Buster/TNT[59] and manual building with Coot[60]. The solvent content of protein crystals was high (68.5% and 69.8% for the scIg12 and scIg12 + EF3a structures, respectively), which contributed to increase the B-factors of the solved crystal structures. The Ramachandran plot statistics for the scIg12 and scIg12 + EF3a structures show that 94% and 99% of residues lie within favored regions, respectively (two and one residues were in disallowed regions, respectively). To refine the structure of scIg12 + EF3a, different approaches were taken to trace the EF-hand loop without success.

Without the loop, the difference map did not show uninterpreted electron density in that area, except that corresponding to a terbium ion that was confirmed by the anomalous density map (and was found close to the theoretical design position; Fig. 4I). Attempts to manually build the EF-hand loop backbone based on the design model led to worse density maps and refinement statistics. We also generated composite omit maps using simulated annealing, but no density was found in that area either. Overall, the challenge of tracing the loop backbone suggests relatively high flexibility, and hence, we concluded it is more appropriate to display the structure without the loop. Final models were evaluated at the wwwPDB Validation Service (https://validate-rcsb-1.wwpdb.org/validservice).

## Tb³⁺ binding luminescence measurements
Time-resolved luminescence emission spectra and intensities were measured on a Synergy H1 hybrid multi-mode reader (BioTek) in flat bottom, black polystyrene, 96-well half-area microplates (Corning 3694). For Tb³⁺ luminescence experiments, protein samples were prepared in 20 mM Tris, 50 mM NaCl, pH 7.4. A TbCl₃ (Sigma-Aldrich, 451304-1G) stock solution was prepared in the same protein buffer. Time-resolved luminescence intensities were measured using excitation wavelength $\lambda_{ex} = 280$ nm and emission wavelength $\lambda_{em} = 544$ nm with a delay of 300 μs, 1 ms collection time and 100 readings per data point. Time-resolved luminescence emission spectra between 500 nm and 600 nm were collected in 5 nm increments. For Tb³⁺ titrations, samples were incubated for 1 h and the collected time-resolved luminescence emission intensities at $\lambda_{em} = 544$ nm were normalized to obtain protein bound fractions. The normalized data was fit to a 1:1 binding model using non-linear least squares regression. Ca²⁺ binding was probed by measuring the decrease of time-resolved terbium luminescence emission intensity at $\lambda_{em} = 544$ nm. For scIg12 + EF3a, CaCl₂ was titrated in the same protein sample buffer into 10 μM protein and 10 μM Tb³⁺. The Ca²⁺ binding affinity constant ($K_d(Ca^{2+})$) was calculated from the apparent $K_d$ ($K_{app}(Ca^{2+})$) using the following equation: $K_d(Ca^{2+}) = K_{app}(Ca^{2+})/(1 + [Tb^{3+}]/K_d(Tb^{3+}))$; where $K_d(Tb^{3+})$ is the terbium-binding affinity constant calculated from Tb³⁺ titrations.

## Visualization and graphical representation
Protein visualization and image rendering were performed with PyMOL[61]. Data analyses were performed with numpy and matplotlib[62].

## Reporting summary
Further information on research design is available in the Nature Portfolio Reporting Summary linked to this article.

## Data availability
Coordinates and structure factors have been deposited in the Research Collaboratory for Structural Bioinformatics Protein Data Bank with the accession codes 8BL3 (design scIg12), 8BL6 (design scIg12 + EF3a). The designed protein structures with high-confident AlphaFold2 predictions, including those experimentally tested, are provided in Supplementary Data 1 along with their AlphaFold2 predictions (five models), and their corresponding sequences are provided in Supplementary Tables 8–10. X-ray crystallography statistics are provided as Supplementary Table 4. The AlphaFold Protein Structure database used for structural analysis is freely available (https://alphafold.ebi.ac.uk). Previously published structures from the Protein Data Bank that were referenced throughout the manuscript are freely available: 5YD3, 7SKO, 7SKP, 4JOX, 1NKF. Other data are available from the corresponding author upon request. Source data are provided with this paper.

## Code availability
The Rosetta macromolecular modeling suite (http://www.rosettacommons.org) is freely available to academic and non-commercial users. The local installation of Colabfold is freely

available at https://github.com/YoshitakaMo/localcolabfold. Other software used in this paper for protein structure design and analysis are freely available: ProteinMPNN (https://github.com/dauparas/ProteinMPNN.git), deep-network hallucination with AlphaFold2 (https://github.com/bwicky/oligomer_hallucination), and Foldseek (https://search.foldseek.com). Custom python scripts for design and analysis are provided in Supplementary Data 1.

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

## Acknowledgements

We are grateful to the joint IBMB/IRB Automated Crystallography Platform and the Protein Purification Service for assistance during SEC-MALS, purification procedures, and crystallization experiments. We thank F.Xavier Gomis-Rüth and Joan Pous for assistance during crystallographic refinement. We also thank Irene Fernandez (CCiTUB) for collecting mass spectrometry data. The authors would further like to thank the ALBA synchrotrons for beamtime allocation and the respective beamline staff for assistance during diffraction data collection. We acknowledge computing resources provided by the Galicia Supercomputing Center (CESGA), and the Red Española de Supercomputación (grants BCV-2021-1-0014 and BCV-2021-3-0010). This research was supported by grants from the Spanish Ministry of Science and Innovation (RYC2018-025295-I, EUR2020-112164 and PID2020-120098GA-I00). J.R.T. was supported by an EMBO postdoctoral fellowship (under grant agreement ALTF 145-2021). Further funding was obtained by AGAUR through grant no. 2021SGR00423.

## Author contributions

J.R.T., M.N., and E.M. designed the research. J.R.T. carried out design calculations, molecular simulations, and protein structure analysis. M.N. expressed, purified, and characterized the proteins with biochemical assays. M.N. crystallized and solved the structure of designed proteins. E.M. carried out design calculations and protein structure analysis. J.R.T, M.N., and E.M. prepared the manuscript.

## Competing interests

J.R.T., M.N., and E.M. have filed a US non-provisional patent application with serial number US 18/177,367 on discoveries described in this manuscript.
