## [Peer Review File · Nature Communications]

Reviewers' Comments:

Reviewer #1:

Remarks to the Author:

The authors present de novo designed single chain immunoglobulin dimers with novel interfaces, which are different from classical ones in Fv fragments, and these proteins feature a novel two-domain immunoglobulin architecture with optimized folding stability. These computationally designed scaffolds, composed of 12 or 14 beta strands, are of extreme thermal and chemical stability, as they show the same degree of secondary structure at temperatures up to 90°C and withstand the effect of the denaturing agent guanidinium chloride at 7 M concentration. They also confirm the computational design of molecules (quite large for de novo design) with X-ray crystal structures.

Additionally, they show that beta-hairpin sites connecting the monomers can accommodate binding loops at multiple positions, either individually or in combination. These motifs have as well been predicted computationally and then confirmed by biophysical characterization and binding assays.

Finally, they demonstrate the robustness of the de novo scaffolds by showing that the domain components, interfaces between the domains, and positioning of functional sites can be highly diversified.

This work is highly significant for the field and uses modern methodology to realize the potential of design of hyperstable (functionalized) immunoglobulin entities de novo.

For computational design of parallelly oriented dimers, they have started with 7-stranded de novo designed monomers, which showed edge-to-edge homodimeric interfaces, and optimized an alpha-helix design flanked with beta-alpha and alpha-beta loops, using Rosetta sequence design. Folding of the designs was assessed using Alpha Fold 2, and several designs with highly confident predictions matched closely the designed structures.

For the antiparallel design, the 7th strand was removed and short loops introduced for bridging, which resulted in a 12-stranded beta sandwich build up of 2 6-stranded immunoglobulin monomers. Three candidates resulting from the antiparallel and one from the parallel group of dimers were found to be well expressed, monomeric in size exclusion chromatography, and extremely thermostable and resistant to chemical denaturation, judging by their far UV-CD profile at 95°C and retained maximum ellipticity with 7 M guanidinium chloride. They also solved the crystal structure of the molecule of antiparallel design, which agreed very well with the computational model.

They also investigated if alternatively, face-to-face orientation could be designed with similar methods. Best designs were studied by Rosetta ab initio folding simulations, however the energy landscapes were less funnelled than the ones found for edge-to-edge design, and there were lower energy gaps between the competing conformation, which tend to have more open hydrophobic interfaces.

Next, they attempted to introduce functional loops at the interface between domains, and grafted a 12-meric EF-hand calcium binding motif extended with linkers to 15-21 amino acids in total. The designs were studied with molecular dynamics simulations and Alpha Fold2, and for 4 designs the predictions agreed with the designed structure. On those, MD simulations were carried out and from 3 most rigid designs, the one with the Alpha fold 2 prediction with the highest confidence was chosen for experimental characterization. It exhibited the same outstanding biophysical properties as the parental molecule and the grafted motif was functional. Following, additional loop insertions were tested at different beta-hairpin positions and combined in pairs. After evaluation of 9 designs with Alpha fold 2, three featuring grafted loops on the opposite sides and one with two loops at the same side were chosen for experimental characterization. Again, they were well expressed, monomeric and highly thermostable, and the ability of simultaneous ligand binding by such scaffolds was demonstrated.

Further, the structural space of extended immunoglobulin scaffolds was explored using deep-learning hallucination with Alpha fold 2 network, by combining in pairs de novo immunoglobulin monomers and hallucinating the 4 terminal strands involved in the edge-to-edge interface. The authors conclude that the domain pairs can be combined and fused through a variety of mechanisms, the number of strands can be tuned, extensions of terminal strands in the linker regions can alter the strand pairing, and curvature of edge strands can shorten the distance between termini of domains that would otherwise not be compatible for fusion.

To conclude, the authors have successfully designed immunoglobulin framework using edge-to-

edge interfaces, and the resulting proteins were hyperstable scaffolds. The binding loop insertion and multiple insertions further evidence their robustness. It will be very interesting to see these single-chain immunoglobulin dimers to harbor binding sites for larger antigens, derived either by grafting, computational design or library screening, and the designs with 2 binding loops on the opposite side of the dimer are an interesting starting point for potential derivatization into stable small-sized bispecific molecules.

I would have only minor questions and remarks to the authors:

Protein yields from E.coli expression should be indicated.

It is indicated that SEC-MALS was measured, but only SEC data are presented for now. It would also be interesting to know up to which concentrations the constructs were soluble and monomeric. The SEC elution profiles in 5c do not quite overlap- what is the information from MALS on these proteins?

Proteins were purified using Ni-affinity chromatography and size exclusion chromatography- it would be interesting to know what was the percentage of monomer before preparative SEC step. Figure 2 a,e: unfortunately, the presentation is not that great with only green and blue visible, and for someone not acquainted with alphafold scores the importance of red and blue in pLDDT is not so obvious-please amend the description where "red" appears. The same comment for Figure 3b. Figure 2 c,g: please indicate the elution of marker standard protein (with dots, or similar). The same for Figure 4g.

Figure 3 e, f: the inset labels are really tiny

Line 378: E.coli, and all in italics

Figure 4 i-k: this is luminescence

Figure 5c to e: it should be clear which data corresponds to a certain mutant.

Line 477: maybe largest monomeric all-beta?

Figure S2: panels should be labelled with a, b, c...

Reviewer #3:

Remarks to the Author:

This paper aims to develop stable and robust antibody-like artificial proteins, by creating single-chain dimers of previously created de novo designed Ig fold structures. The authors proposed principles, in which the single-chain Ig-fold dimers can be created by utilizing edge-to-edge b-strand interactions of the Ig fold structures and fusing these structures by a short and rigid linker. Based on the principles, the authors succeeded in designing the single chain dimers with super high stability. The authors also demonstrated the ability to introduce functional sites into the multiple loops of scIg12 by incorporating an EF-hand motif into one or two loop regions. Furthermore, the authors showed that the designed single-chain dimers can be diversified in shape by using a deep-learning based protein design method.

The paper is appreciated for successfully designing single-chain dimers based on Ig fold structures that the author has previously designed de novo and for demonstrating the ability to introduce functional EF-hand motifs into multiple beta-hairpins. My major concern is that it is unclear whether the claimed "robustness" and "versatility" have been demonstrated. While antibodies can change the sequence of their variable regions in various way, the authors only tested only EF-hand motif. It is unclear how diverse the sequences that can be introduced into the beta-hairpins in the designed structures. As for versatility, it is unclear how this was proven experimentally.

Minor points:

- In Fig.1, while the figure shows four possible single-chain dimers arrangements for the 7-stranded and 6-stranded Ig domain, there would be other ones? For example, for the top-left panel of the 7-stranded Ig domain, there is a pattern where the red and blue strands are parallel. Why were only four possibilities shown? All patterns should be described.
- Line 117: "Fig. 1b left" should be "left Fig. 1c left".
- Line 138-142, it was difficult to understand (actually, I was not able to understand) the sentences. These are needs to be revised.
- Line 167-168 and Figure S1, for my understanding, the ba-loop is the one connecting a beta-strand to an alpha-helix, and the ab-loop is the one connecting a alpha-helix to a beta-strand?

These descriptions are opposite in the manuscript.

- Lines 177-178, GBA-H14-BAB and BA-H14-BAB should be BAB-H14-GBA and BAB-H15-BA, respectively.
- Line 223 states "novel protein topologies", but TM-scores are relatively high. Are they really novel topologies?
- Regarding 6), in Fig. S4 and S5, instead of overlaying structures, it would be helpful to display structures side-by-side and number the strands in order to clearly show the topology differences.
- The graphs in Fig 5. c, d, and e are hard to see because the lines overlap, and the colors are similar with each other. These are needs to be revised.
- Was an X-ray crystallography analysis performed on scIg 14b? More explanation is required.

Reviewer #4:

Remarks to the Author:

This manuscript presents a computational based protein design project that endeavors to fabricate a family of highly stable scaffolds that can be utilized to "display" a variety of binding loops that are positioned to be functionally active. The authors contend that there is a need for such scaffolds because of perceived limitations of natural scaffolds like antibodies, nanobodies and the like. The constituted constructs are based on designs involving extended dimeric Ig β -sandwiches with variable numbers of strands. It was proposed that these single-chain (sc) dimer Ig scaffolds would be hyper-stable and provide platforms for introducing functional binding loops into their structures. Several of the of the sc dimer constructs were expressed in Ecoli and biophysically characterized showing that they were indeed highly stable entities. Further, X-ray structure analyses were performed which demonstrated that the extended β -sheet components of the designed structures are closely comparable to the experimental determined ones. The single proof of principle example provided in the manuscript is the substitution of an EF-hand motif into the β -hairpin bridge connecting domains of the single-chain dimer. They show that inserted EF-Hand can bind terbium in a concentration dependent manner. However, one might argue that this EF-Hand example is not really a proof of concept for the cases involving loop recognition that promote protein-protein interactions of the type proposed here.

This reviewer is not an expert in protein design, so I will leave the assessment of that part of the work to other reviewers. The biophysical analyses appear generally well done; however, the X-ray structure analysis seems to be a work in progress. It is apparent from the X-ray structure that the inserted EF-Hand motif has significant inherent flexibility, making it impossible to trace its chain. This suggests that inserted motifs, loops, etc., will be displayed structurally disconnected from the framework scaffold in which it is inserted. This is quite different than the situation in antibodies where myriad examples exist from humanization efforts showing the framework residues play a large role in the loop's presentation. Furthermore, only a small fraction of antibodies employ only a single CDR loop for their antigen binding, rather there is generally highly coordinated participation of multiple CDRs, either directly or indirectly. It is not clear how the rigid framework of the sc Ig dimers could accommodate this type of thing, so it is important for the authors to provide some additional examples to support their supposition that these molecules could have antibody like properties (or even can display loops without loss of binding affinity). Otherwise, they need to step back their claims about the potential utility of these types of molecules. Right now, the described potential functionality comes across as a lot of hype with no real proof of concept for reproducing protein-protein molecular recognition properties. What is clear is that this is far from a plug and play strategy for introducing motifs/loops into these scaffolds to mimic functional properties. Under the best of circumstances, prospective practitioners would be faced with fairly sophisticated modeling with little proof of a successful payoff in the end. The hard work of assessing whether this is a viable concept for real world applications is left to the user.

Below are specific comments:

- 1). Considering the stability of the Ig dimer scaffold, it is surprising that the B-factors are so high. This type of thing is expected for membrane proteins, not stable soluble ones. A back of the envelope calculation suggests that the problem may be attributed to the high solvent content of the crystals? The authors should provide this information and comment.
- 2). Particularly disturbing is the high B-factor for the Terbium ion (400 \AA^2). Basically, this is indicating that the Terbium is misplaced or not there. The anomalous difference density indicates

that it is there in the crystal, although the occupancy could be low. Additionally, the position of the Terbium in Fig. 4I (orange sphere) doesn't align well with the anomalous density (at least in the view provided). It would make more sense to place the Terbium into the density and build the loop around that. The coordination geometry of ions incapsulated in EF-Hands is well established (and utilized in the designs of scig12+EF). Superimposing this structure with the right stereochemistry centered around the ion would provide a good starting point to build the loop. Although local flexibility might still be problematic, given the protein design prowess of the authors, it should be straightforward to predict where this loop has to be using the ion position as a guidepost.

3). Several potential attributes of sc dimers were mentioned, but there are obvious shortcomings, as well if there are thoughts of "replacing" antibodies for clinical applications- half life, immunogenicity, cross reactivity, developability and so forth. Sure, some of this can be engineered into the molecules, but at a cost of time and efficiency.

In summary, the manuscript describes what appears to be a successful design/engineering endeavor to fabricate a family of highly stable β -sheet structures. However, the premise that these structures can be exploited in a practical way remains to be proven. It is essential to actually show that paratopes grafted into these structures perform in a similar fashion as the loops of the protein they are derived from.

Changes made in response to the comments of the Reviewers

In the following sections, the comments of the Reviewers are written in italics. Our responses to each of the items are presented under their comments in blue font. Excerpts from the text are written in smaller blue font and indented.

Reviewer 1

The authors present de novo designed single chain immunoglobulin dimers with novel interfaces, which are different from classical ones in Fv fragments, and these proteins feature a novel two-domain immunoglobulin architecture with optimized folding stability. These computationally designed scaffolds, composed of 12 or 14 beta strands, are of extreme thermal and chemical stability, as they show the same degree of secondary structure at temperatures up to 90°C and withstand the effect of the denaturing agent guanidinium chloride at 7 M concentration. They also confirm the computational design of molecules (quite large for de novo design) with X-ray crystal structures.

Additionally, they show that beta-hairpin sites connecting the monomers can accommodate binding loops at multiple positions, either individually or in combination. These motifs have as well been predicted computationally and then confirmed by biophysical characterization and binding assays.

Finally, they demonstrate the robustness of the de novo scaffolds by showing that the domain components, interfaces between the domains, and positioning of functional sites can be highly diversified.

This work is highly significant for the field and uses modern methodology to realize the potential of design of hyperstable (functionalized) immunoglobulin entities de novo.

We thank the Reviewer for these positive comments

For computational design of parallelly oriented dimers, they have started with 7-stranded de novo designed monomers, which showed edge-to-edge homodimeric interfaces, and optimized an alpha-helix design flanked with beta-alpha and alpha-beta loops, using Rosetta sequence design. Folding of the designs was assessed using Alpha Fold 2, and several designs with highly confident predictions matched closely the designed structures.

For the antiparallel design, the 7th strand was removed and short loops introduced for bridging, which resulted in a 12-stranded beta sandwich build up of 2 6-stranded immunoglobulin monomers. Three candidates resulting from the antiparallel and one from the parallel group of dimers were found to be well expressed, monomeric in size exclusion chromatography, and extremely thermostable and resistant to chemical denaturation, judging by their far UV-CD profile at 95°C and retained maximum ellipticity with 7 M guanidinium chloride. They also solved the crystal structure of the molecule of antiparallel design, which agreed very well with the computational model.

They also investigated if alternatively, face-to-face orientation could be designed with similar methods. Best designs were studied by Rosetta ab initio folding simulations, however the energy landscapes were less funnelled than the ones found for edge-to-edge design, and there were lower energy gaps between the competing conformation, which tend to have more open hydrophobic interfaces.

Next, they attempted to introduce functional loops at the interface between domains, and grafted a 12-meric EF-hand calcium binding motif extended with linkers to 15-21 amino acids in total. The designs were studied with molecular dynamics simulations and Alpha Fold2, and for 4 designs the predictions agreed with the designed structure. On those, MD simulations were carried out and from 3 most rigid designs, the one with the Alpha fold 2 prediction with the highest confidence was chosen for

experimental characterization. It exhibited the same outstanding biophysical properties as the parental molecule and the grafted motif was functional. Following, additional loop insertions were tested at different beta-hairpin positions and combined in pairs. After evaluation of 9 designs with Alpha fold 2, three featuring grafted loops on the opposite sides and one with two loops at the same side were chosen for experimental characterization. Again, they were well expressed, monomeric and highly thermostable, and the ability of simultaneous ligand binding by such scaffolds was demonstrated. Further, the structural space of extended immunoglobulin scaffolds was explored using deep-learning hallucination with Alpha fold 2 network, by combining in pairs de novo immunoglobulin monomers and hallucinating the 4 terminal strands involved in the edge-to-edge interface. The authors conclude that the domain pairs can be combined and fused through a variety of mechanisms, the number of strands can be tuned, extensions of terminal strands in the linker regions can alter the strand pairing, and curvature of edge strands can shorten the distance between termini of domains that would otherwise not be compatible for fusion. To conclude, the authors have successfully designed immunoglobulin framework using edge-to-edge interfaces, and the resulting proteins were hyperstable scaffolds. The binding loop insertion and multiple insertions further evidence their robustness. It will be very interesting to see these single-chain immunoglobulin dimers to harbor binding sites for larger antigens, derived either by grafting, computational design or library screening, and the designs with 2 binding loops on the opposite side of the dimer are an interesting starting point for potential derivatization into stable small-sized bispecific molecules.

We thank the Reviewer for clearly summarizing the main results of the manuscript.

I would have only minor questions and remarks to the authors:

Protein yields from E.coli expression should be indicated.

We have now provided protein yields in the Methods section “Recombinant expression and purification of the designed proteins” with the following sentence:

Overall, we have obtained between 2 and 6 mg of purified protein per liter of cell culture without optimization of expression conditions.

Since protein yields were enough for our experimental characterization of the designed proteins we did not perform any optimization of expression conditions.

It is indicated that SEC-MALS was measured, but only SEC data are presented for now. It would also be interesting to know up to which concentrations the constructs were soluble and monomeric. The SEC elution profiles in 5c do not quite overlap- what is the information from MALS on these proteins?

SEC chromatograms obtained right after Ni affinity purification are now provided in Supplementary Figures 3c,g and 4. We have now provided the SEC-MALS chromatograms (obtained from the SEC fractions with elution volumes as expected for the molecular size of the monomeric state) in the main text figures (Figs. 2c,g, 4g and 5e). We have also added the molecular weights estimated from SEC-MALS in Supplementary Table 2. All figure captions have been updated accordingly. From this data proteins are predominantly monomeric. Although we have not analyzed explicitly the maximum concentration at which proteins remain monomeric, the crystal structures do not show a tendency to form oligomeric interactions.

After rearranging Fig. 5, panel c becomes e. The elution profiles in 5e not only correspond to the 4 bifunctional designs, but also to scIg12 and scIg12+EF3a. The figure caption reads as:

e, SEC-MALS chromatograms of the purified single-chain dimers with two functional loops (colored as in (d)) in comparison to scIg12+EF3a (gray) and their parental scIg12 (black)

Bifunctional designs are expected to elute earlier than scIg12+EF3a, which in turn elutes earlier than scIg12. The three design series follow the trend as expected from their molecular size differences (molecular size increases by adding functional loops) The elution profiles of the 4 bifunctional designs are highly similar, also as expected. They have minor differences in molecular shape, due to different loop location and structure, that can slightly affect the elution volume.

Proteins were purified using Ni-affinity chromatography and size exclusion chromatography– it would be interesting to know what was the percentage of monomer before preparative SEC step.

It is worth clarifying that after Ni affinity purification, we have performed size-exclusion chromatography purification. Then, for the peak eluting as expected based on the protein molecular size we have collected fractions and performed SEC-MALS to estimate the molecular weight. Overall, for most proteins the SEC chromatograms are monodisperse and have a dominant peak corresponding to the monomeric state (based on SEC-MALS data).

Figure 2 a,e: unfortunately, the presentation is not that great with only green and blue visible, and for someone not acquainted with alphafold scores the importance of red and blue in pLDDT is not so obvious-please amend the description where “red” appears. The same comment for Figure 3b.

We thank the Reviewer for noting this. We have been particularly interested in stressing the importance of high- (in blue) and low-confidence (in red) regions in the AlphaFold models, as this relates to the quality of the designs. For a clearer visualization, we have replaced overlaid models by side-to-side models in Fig.2a,e and Fig.3b (left). We have also updated the captions regarding the coloring scheme according to pLDDT values.

Figure 2 c,g: please indicate the elution of marker standard protein (with dots, or similar). The same for Figure 4g.

To estimate the molecular weight of the purified proteins we have only relied on SEC-MALS analysis, so we have not used any standard in the SEC. Now we have added a Supplementary Table 2 that includes the molecular weights estimated from SEC-MALS, in comparison to the theoretical ones.

Figure 3 e, f: the inset labels are really tiny

We understand that the Reviewer refers to panels e and f from Figure 4, as there are no such panels in figure 3. We agree that the inset labels of Fig. 4e and 4f are difficult to read. We have removed them and provided a clear specification of each data series in the figure caption.

Line 378: E.coli, and all in italics

This has been fixed.

Figure 4 i-k: this is luminescence

Thank you for spotting this out. We have also updated Fig.5f,g for consistency and the caption accordingly.

Figure 5c to e: it should be clear which data corresponds to a certain mutant.

We have now changed the coloring scheme and updated the caption.

Line 477: maybe largest monomeric all-beta?

We fully agree and have added this specification.

Figure S2: panels should be labelled with a, b, c...

We have now labeled the panels as suggested, and updated the caption accordingly.

Reviewer 2

This paper aims to develop stable and robust antibody-like artificial proteins, by creating single-chain dimers of previously created de novo designed Ig fold structures. The authors proposed principles, in which the single-chain Ig-fold dimers can be created by utilizing edge-to-edge b-strand interactions of the Ig fold structures and fusing these structures by a short and rigid linker. Based on the principles, the authors succeeded in designing the single chain dimers with super high stability. The authors also demonstrated the ability to introduce functional sites into the multiple loops of scIg12 by incorporating an EF-hand motif into one or two loop regions. Furthermore, the authors showed that the designed single-chain dimers can be diversified in shape by using a deep-learning based protein design method.

The paper is appreciated for successfully designing single-chain dimers based on Ig fold structures that the author *has previously designed de novo and for demonstrating the ability to introduce functional EF-hand motifs into multiple beta-hairpins.*

We thank the Reviewer for the positive assessment.

My major concern is that it is unclear whether the claimed “robustness” and “versatility” have been demonstrated. While antibodies can change the sequence of their variable regions in various way, the authors only tested only EF-hand motif. It is unclear how diverse the sequences that can be introduced into the beta-hairpins in the designed structures. As for versatility, it is unclear how this was proven experimentally.

Several of our results support the robustness of the single-chain immunoglobulin dimers to tolerate a diverse range of functional loop insertions. For improved clarity, we have added some sentences throughout the text and a new figure panel (Fig. 5b) showing the diversity of designed loops. It is worth clarifying that the designed functional loops not only involve the EF-hand sequence motif but also the N- and C-terminal linkers that correctly integrate it into the scaffold. For this reason, the functional loops were designed in a wide range of lengths (between 15 and 23 amino acids). Overall, the designed loops varied in terms of length, sequence and amino acid composition, and were incorporated in multiple β -hairpin positions. For the sake of clarity, we have now added Fig. 5b displaying a multiple sequence alignment of all the designed functional loops that passed our structure quality filters, which clearly shows the loop diversity of the designs. The experimentally tested proteins were well-expressed, monomeric and highly stable, regardless of the loop sequence and b-hairpin location. We understand that these are strong indicators of the robustness of the scIg12 scaffold. Additionally, the ability to tolerate pairs of functional loops in multiple combinations further demonstrates such robustness. Finally, the crystal structures of scIg12 and scIg12+EF3a were nearly identical in the scaffold structure, pointing also to the robustness of the structure to long loop (15 amino acids) insertions. Some of these ideas were already conveyed in the first paragraph of the Discussion, but for the sake of more clarity, we have also included the following paragraph in the Discussion section:

The robustness of the edge-to-edge single-chain Ig dimers for scaffolding multiple loops opens exciting possibilities in several directions. The designed binding loops (experimentally tested and/or *in silico* validated) were diverse in terms of sequence, length, amino acid composition and scaffold position. Across three independent β -hairpin sites, the 12-residue EF-hand motif was correctly integrated in loops varying in length (ranging between 15 and 23 residues), and combining amino acids of all types – i.e., polar (S, T, Q, N), charged (K, E, D), aliphatic (A, I, L), aromatic (Y), and turn-forming (G, P) – as shown in Fig. 5b. Such length range is comparable, or larger, than the average length typically found in CDRs of

antibodies^{41,42}. Also, amino acid diversity in loop sequences is particularly relevant for designing protein-protein interactions⁴², which require combining polar and hydrophobic interactions, and amino acids controlling loop conformational stability. The designed loops were structurally connected to the rest of the scaffold to a varying degree (Supplementary Fig. 13), which also influences the shape of epitopes that can be targeted. (e.g., protruding loops are in general better suited for targeting concave epitopes, as seen in nanobodies inhibiting enzyme activities⁴). Our results indicate that single-chain Ig dimeric scaffolds have the robustness to accommodate a wide range of loops in multiple sites (both individually and in combination), which holds promise for designing protein binding paratopes in a similar spirit to antibodies or nanobodies. In this regard, this work represents a stepping stone towards de novo designing antibody-like protein binding scaffolds. Based on our encouraging results, it will be interesting in future work to design binding loops for protein targets of interest through high-throughput screening of loop libraries^{43,44} (naive or computationally focused) or computational design. Indeed, recent advances in deep-learning should facilitate the design of entirely new binding loops⁴⁵, without the need of grafting existing motifs and solely based on the target epitope structure. Furthermore, the tunability of the scaffold structure will be advantageous to provide loop conformational stability, and make additional contacts with protein targets (as in nanobodies⁴² or monobodies³).

On the other hand, we agree with the Reviewer that versatility has not been proven experimentally, but single-chain Ig dimer proteins have structural features that are well-suited for other applications beyond loop scaffolding; which would make them very versatile proteins. Definitely it will be worth exploring these possibilities in future work, and we believe this manuscript lays the groundwork for it. We have now reviewed the text to clarify that we refer to the versatile potential of these scaffolds, rather than versatility demonstrated by the results. We have rephrased some parts of this paragraph in the Discussion section:

Single-chain Ig dimers may be suitable for diverse applications beyond binding single protein targets. The robustness of these scaffolds to simultaneously display loops in the two distal sides could be useful for creating compact, stable and single-chain bispecific formats, which are particularly useful for co-localizing proteins of interest in a broad range of applications. Beyond loop scaffolding, single-chain Ig dimers could also shape binding pockets for small-molecules or metal ions on the top (or bottom) of the interface between domains, where the structure of the scaffold and surrounding loops could be readily diversified to form cavities matching binding site geometries (the crystal structures of scIg12 and scIg12+EF3a reveal a crystallization component binding to the top of the interface that is surrounded by several loops). The high stability of these scaffolds should ease the incorporation of cavity-creating mutations, as described for other de novo proteins⁴⁶. Also, their extended β -sheet structures show potential as traps for extended peptides, as recently shown with other protein scaffolds recognizing amyloidogenic peptide segments through extended β -sheets leaving unpaired strands in the middle⁴⁷. The versatile potential of single-chain Ig dimers opens exciting avenues worth exploring in future work.

Minor points:

- In Fig.1, while the figure shows four possible single-chain dimers arrangements for the 7-stranded and 6-stranded Ig domain, there would be other ones? For example, for the top-left panel of the 7-stranded Ig domain, there is a pattern where the red and blue strands are parallel. Why were only four possibilities shown? All patterns should be described.

As we wanted to assess the possibility of designing single-chain dimers by sequence-local fusion of 2 de novo immunoglobulins, we identified that one of the requirements was to have the C-terminal strand of the first monomer and N-terminal strand of the second one paired on the same β -sheet layer. This is one of the structural requirements to optimize folding stability of two-layer, edge-to-edge single-chain dimers of Ig domains that we describe in the second paragraph of the Results section. For this reason,

in Fig. 1 we have only represented the four arrangements matching these requirements. We have updated the caption to clarify this. Alternative arrangements would involve non-local b-arch connections that would be in principle less optimal in terms of folding.

- Line 117: “Fig. 1b left” should be “left Fig. 1c left”.

This has been fixed.

- Line 138-142, it was difficult to understand (actually, I was not able to understand) the sentences. These are needs to be revised.

We have now rewritten the sentence to better clarify this point in the manuscript, as follows:

Also, the two-layer hydrogen bond pairing is likely more optimal if the global β -sandwich geometries of the two Ig domains – in terms of distance separation and rotations between the two β -sheets of each domain – are similar; otherwise, strand pairing in one layer would be incompatible with a second pairing in the other layer (see Supplementary Fig. 1).

For further clarity, we have also added a new figure in Supplementary Information (Supplementary Fig. 1) that illustrates this idea.

- Line 167-168 and Figure S1, for my understanding, the ba-loop is the one connecting a beta-strand to an alpha-helix, and the ab-loop is the one connecting a alpha-helix to a beta-strand? These descriptions are opposite in the manuscript.

The Reviewer is right. We have recalculated the blueprints and ABEGOs for $\beta\alpha$ and $\alpha\beta$ loops, which are BAB-H14-GBA, BAB-H14-GB and BAB-H14-BA for scIg14a, scIg14b and scIg14c, respectively. We have corrected this mistake throughout the manuscript (including the Methods section) and updated the Supplementary Figure (now Supplementary Fig. 2)

- Lines 177-178, GBA-H14-BAB and BA-H14-BAB should be BAB-H14-GBA and BAB-H15-BA, respectively.

This has been fixed.

- Line 223 states “novel protein topologies”, but TM-scores are relatively high. Are they really novel topologies?

We thank the Reviewer for bringing up this point. We have taken the opportunity to revisit our topology searches with the newest version of the Foldseek server and the updates to the PDB and AlphaFold databases. Our new searches suggest that the β -sheet arrangement of our scaffolds is a protein topology very hard to find in nature. The top-ranked hits are β -sandwiches with a lower number of β -strands and distinct strand pairing organization. We have updated the old Supplementary Figures 4 and 5 to Supplementary Figures 6 and 7, which better reflect these findings. Also, we have included a new section in the manuscript that further elaborates on this:

Structural analogs of single-chain immunoglobulin dimers in nature

Although β -sandwiches are ubiquitous in nature, the complex β -sheet arrangement of two-layer single-chain immunoglobulin dimers leads to protein topologies – i.e., number of strands, their connections and pairings (parallel or antiparallel) – that are hard to find in nature. We searched for structural analogs of the scIg12 and scIg14 designs in the PDB and the AlphaFold Protein Structure Database³² (AlphaFold DB version Nov 2022) using the Foldseek server³³. The top hits found in the PDB were β -sandwiches with a lower number of β -strands and different strand pairing organization – the top hit is a 9-stranded β -sandwich (PDB code: 4JOX) with a low TM-score³⁴ of 0.52 (Supplementary Fig. 6b). In the AlphaFold DB, the top-hit model retrieved by Foldseek was an 8-stranded β -sandwich of low confidence (pLDDT 67), also with a TM-score of 0.52 (Supplementary Fig. 6c). We manually inspected other hits from the AlphaFold DB and only found three models with topologies partially similar to our designs and connecting two Ig domains in alternative ways (Supplementary Fig. 7). First, a multidomain protein (Uniprot accession number K0ESF4) containing a single-chain Ig dimer (predicted with high confidence; pLDDT of 96), where both domains have edge-to-edge pairing only in one layer and are fused through a β -arch connection that rotates one domain with respect to the other (Supplementary Fig. 7a); having a TM-score of 0.51 with our designs. In the second hit (Uniprot accession number P81333), two interacting Ig domains are fused through an extended loop connection that prevents edge-to-edge pairing (Supplementary Fig. 7b). The third hit (Uniprot accession number A0A0S6VQQ4) is the closest analog to the scIg12 design (TM-score of 0.69), and corresponds to a two-layer single-chain Ig dimer (although predicted with low confidence; pLDDT of 69) that inserts a subdomain between two Ig domains, and has suboptimal edge-to-edge pairing due to a register shift in one of the two layers (Supplementary Fig. 7c).

- Regarding 6), in Fig. S4 and S5, instead of overlaying structures, it would be helpful to display structures side-by-side and number the strands in order to clearly show the topology differences.

These figures have been replaced by Supplementary Figures 6 and 7 based on these comments and according to the analysis on structural analogs described above.

- The graphs in Fig 5. c, d, and e are hard to see because the lines overlap, and the colors are similar with each other. These are needs to be revised.

We have now changed the coloring scheme and updated the caption.

- Was an X-ray crystallography analysis performed on scIg 14b? More explanation is required.

Our crystallization efforts on scIg14 designs did not succeed. We then focused our efforts on solving the X-ray structure of scIg12+EF3a to verify the structural recapitulation of the scaffold after functional insertion.

Reviewer 3

*This manuscript presents a computational based protein design project that endeavors to fabricate a family of highly stable scaffolds that can be utilized to “display” a variety of binding loops that are positioned to be functionally active. The authors contend that there is a need for such scaffolds because of perceived limitations of natural scaffolds like antibodies, nanobodies and the like. The constituted constructs are based on designs involving extended dimeric Ig β -sandwiches with variable numbers of strands. It was proposed that these single-chain (sc) dimer Ig scaffolds would be hyper-stable and provide platforms for introducing functional binding loops into their structures. Several of the of the sc dimer constructs were expressed in *E. coli* and biophysically characterized showing that they were indeed highly stable entities. Further, X-ray structure analyses were performed which demonstrated that the extended β -sheet components of the designed structures are closely comparable to the experimental determined ones.*

We thank the Reviewer for taking the time to review our manuscript.

The single proof of principle example provided in the manuscript is the substitution of an EF-hand motif into the β -hairpin bridge connecting domains of the single-chain dimer. They show that inserted EF-Hand can bind terbium in a concentration dependent manner.

The first proof of principle provided in the manuscript is the fusion of two immunoglobulin domains through a terbium-binding loop that integrates an EF-hand motif. However, it is worth clarifying that this is not the only proof of principle. In the Results section entitled “Scaffolding multiple binding loop combinations in single-chain Ig dimers”, and the corresponding Figure 5, we demonstrate the ability of our single-chain dimers to scaffold multiple functional terbium-binding loops across different β -hairpin regions simultaneously. Specifically, in Fig. 5d we show four representative bivalent designs (EF1a+EF3a, EF2e+EF3a, EF3a+EF4 and EF1a+EF4) displaying two functional loops either on the same or opposite sides. All of them were well-expressed and purified in *E. coli* (Fig. 5e), were monomeric, thermostable and bound terbium in a concentration dependent manner (Fig. 5f). Moreover, we show in Fig. 5g that these four constructs (green, blue, pink and orange series) bind terbium through both loops simultaneously as the measured luminescence enhancement represents a 1.7-2.3 fold increase compared to EF3a (dark gray), which only displays a single loop. Overall, we show 5 different functional constructs: 1 construct displaying a single loop (scIg12+EF3a) and 4 constructs displaying simultaneously two binding loops at either opposite sides (scIg12+EF1a+EF3a, scIg12+EF2e+EF3a and scIg12+EF3a+EF4) or same side (scIg12+EF1a+EF4). Although we do not provide crystal structures for bifunctional designs, luminescence binding experiments strongly support correct formation of pairs of binding loops.

However, one might argue that this EF-Hand example is not really a proof of concept for the cases involving loop recognition that promote protein-protein interactions of the type proposed here.

We agree with the Reviewer that our functional loops do not bind proteins, but in our view they provide supporting evidence that our scaffolds fulfill two important requirements for designing protein-protein interactions mediated by loops: robustness for functional loop insertion and structural tunability. It is worth clarifying first that our designed functional loops not only involve the EF-hand sequence motif but also the N- and C-terminal linkers that correctly integrate it into the scaffold. For this reason, the functional loops were designed in a wide range of lengths (between 15 and 23 amino acids). Overall, the designed loops varied in terms of length, sequence and amino acid composition, and were

incorporated in multiple β -hairpin positions. For the sake of clarity, we have now added Fig. 5b displaying a multiple sequence alignment of all the designed functional loops that were validated experimentally or computationally, which clearly shows the loop diversity of the designs. The experimentally tested proteins harboring functional loops were all well-expressed, monomeric, highly stable and active; regardless of the loop sequence and β -hairpin location.

The requirement of robustness is necessary to accommodate long functional loops without stability issues and/or binding affinity loss. In general, the inherent flexibility of long loops can hinder folding and stability of protein scaffolds, but single-chain Ig dimers have been conceived in a way that eases functional loop incorporation based on principles of protein folding stability (see “Design principles for edge-to-edge single-chain immunoglobulin dimer” Results section). In this regard, we show that across 3 different β -hairpin sites our scaffolds can incorporate functional loops ranging from 15 to 23 residues and retaining native binding affinities, alone or in combination. Such loop length range is comparable, or larger, than the average length typically found in CDRs of antibodies (Adolf-Bryfogle, J. *et al.* Nucleic Acids Res. 43, D432–D438). Besides length diversity, our designed loops have sequence diversity, as shown in Fig. 5b, and combine all amino acid types across the designs. This is particularly relevant for designing protein-protein interactions, which require combining polar and hydrophobic interactions, and incorporating amino acids controlling loop conformational stability (Gly and Pro). To clarify this point, besides Fig. 5b, we have included a new paragraph in the discussion that we quote here:

The robustness of the edge-to-edge single-chain Ig dimers for scaffolding multiple loops opens exciting possibilities in several directions. The designed binding loops (experimentally tested and/or *in silico* validated) were diverse in terms of sequence, length, amino acid composition and scaffold position. Across three independent β -hairpin sites, the 12-residue EF-hand motif was correctly integrated in loops varying in length (ranging between 15 and 23 residues), and combining amino acids of all types – i.e., polar (S, T, Q, N), charged (K, E, D), aliphatic (A, I, L), aromatic (Y), and turn-forming (G, P) – as shown in Fig. 5b. Such length range is comparable, or larger, than the average length typically found in CDRs of antibodies^{41,42}. Also, amino acid diversity in loop sequences is particularly relevant for designing protein-protein interactions⁴², which require combining polar and hydrophobic interactions, and amino acids controlling loop conformational stability. The designed loops were structurally connected to the rest of the scaffold to a varying degree (Supplementary Fig. 13), which also influences the shape of epitopes that can be targeted. (e.g., protruding loops are in general better suited for targeting concave epitopes, as seen in nanobodies inhibiting enzyme activities⁴). Our results indicate that single-chain Ig dimeric scaffolds have the robustness to accommodate a wide range of loops in multiple sites (both individually and in combination), which holds promise for designing protein binding paratopes in a similar spirit to antibodies or nanobodies. In this regard, this work represents a stepping stone towards *de novo* designing antibody-like protein binding scaffolds. Based on our encouraging results, it will be interesting in future work to design binding loops for protein targets of interest through high-throughput screening of loop libraries^{43,44} (naïve or computationally focused) or computational design. Indeed, recent advances in deep-learning should facilitate the design of entirely new binding loops⁴⁵, without the need of grafting existing motifs and solely based on the target epitope structure. Furthermore, the tunability of the scaffold structure will be advantageous to provide loop conformational stability, and make additional contacts with protein targets (as in nanobodies⁴² or monobodies³).

The requirement of scaffold structural tunability is relevant for controlling the anchoring positions of functional loops, which in turn modulate the relative orientation between loops cooperatively building a protein binding interface. As shown in the “Structural space of extended immunoglobulin β -sandwich scaffolds” section, and the corresponding Fig. 6, these scaffolds can be readily diversified by combining

distinct Ig domains through different fusions controlling the total number and spatial organization of hairpin sites on which binding loops could be incorporated.

We would like to emphasize that de novo designed protein scaffolds fulfilling these requirements have not been described yet, and this work represents a necessary stepping stone towards the *de novo* design of antibody-like proteins binding protein targets of interest.

This reviewer is not an expert in protein design, so I will leave the assessment of that part of the work to other reviewers. The biophysical analyses appear generally well done; however, the X-ray structure analysis seems to be a work in progress. It is apparent from the X-ray structure that the inserted EF-Hand motif has significant inherent flexibility, making it impossible to trace its chain.

This suggests that inserted motifs, loops, etc., will be displayed structurally disconnected from the framework scaffold in which it is inserted.

This is quite different than the situation in antibodies where myriad examples exist from humanization efforts showing the framework residues play a large role in the loop's presentation.

We agree with the Reviewer that the EF3a binding loop must be flexible, as we did not succeed to trace the loop backbone. However, the anomalous diffraction data supports that the terbium ion is located very close to the designed position. We have emphasized this idea in the Results section:

We identified electron density around the designed metal position that was unambiguously assigned to a Tb³⁺ ion (Fig. 4I, inset), as confirmed by anomalous X-ray scattering data (Supplementary Fig. 12), but the binding loop backbone could not be traced due to missing electron density; pointing to loop flexibility even with bound metal.

We would also like to note that it is not uncommon to find protein structures in the PDB with missing segments, especially on the loops. Crystal structures of antibodies can also be found with missing loops on the paratope. This indicates challenges to experimentally resolve structures with flexible components.

It is important to emphasize that the flexibility of binding loops and their integration into the scaffold can vary significantly from case to case. Here, we have focused on loops integrating a 12-residue EF-hand motif, which has a specific conformation and coordination geometry around metal ions. The designed N- and C-terminal linkers accommodating the EF-hand motif into the scaffold play an important role in controlling part of loop rigidity and separation between the loop binding residues and the scaffold. As shown in Supplementary Figure 13, the designed loops with an *in silico* validation are structurally connected to the rest of the scaffold to a varying degree. We cannot rule out the possibility that some of these loops would exhibit lower flexibility in a crystal structure. We have added the following fragment to the Discussion section:

The designed loops were structurally connected to the rest of the scaffold to a varying degree (Supplementary Fig. 13), which also influences the shape of epitopes that can be targeted. (e.g., protruding loops are in general better suited for targeting concave epitopes, as seen in nanobodies inhibiting enzyme activities⁴).

It is important to note that loops protruding from the scaffold to a varying extent will have different protein binding capabilities. For example, it is well established that nanobodies are better suited than

antibodies for targeting concave epitopes, as in enzyme active sites, through a long protruding H3 loop. Essentially, the optimization of loop conformation and separation from the scaffold will highly depend on the shape of the protein target (whether it is flat, concave, convex, or rugged). However, this aspect will be explored in future work. We believe that computational methods for designing completely new loops specifically tailored for epitopes of interest hold promise and could overcome limitations associated with grafting existing CDRs onto our novel scaffolds. This idea has been emphasized in the Discussion section.

Furthermore, only a small fraction of antibodies employ only a single CDR loop for their antigen binding, rather there is generally highly coordinated participation of multiple CDRs, either directly or indirectly. It is not clear how the rigid framework of the sc Ig dimers could accommodate this type of thing, so it is important for the authors to provide some additional examples to support their supposition that these molecules could have antibody like properties (or even can display loops without loss of binding affinity). Otherwise, they need to step back their claims about the potential utility of these types of molecules.

We agree with the Reviewer that to build antigen binding interfaces, as in antibodies or nanobodies, it is necessary to present multiple loops simultaneously. For this reason, as a proof of concept, we have also assessed the ability of our scaffolds to display two loops simultaneously varying in terms of sequence, structure and position in the scaffold (as shown in Fig. 5). All such bivalent designs had similar binding affinities, with both loops simultaneously binding terbium; and these affinities were in turn very similar to that of the monovalent design (scIg12+EF3a). Therefore, the scaffold tolerated a diverse range of functional loop insertions without loss of binding affinity.

Besides such robustness for loop scaffolding, structural tunability is an important requirement for building protein-protein interactions coordinated by multiple loops. As described in the “*Structural space of extended immunoglobulin β -sandwich scaffolds*” Results section, our single-chain dimers exhibit a modularity that allows to readily diversify their structure. Indeed, we illustrate that this modularity offers the possibility of tailoring the number, positioning and relative orientation of binding loops by combining a broad range of Ig domain pairs, either equal or not in terms of structure and/or number of strands. This might be of particular relevance for the design of binding paratopes of high structural diversity and that may not be limited to targeting concave or convex epitopes. For example, one of the bifunctional designs displays two functional loops on the same side of the scaffold. Similarly, protein binding loops could be displayed on the same side to cooperatively build a protein recognition interface. We would like to highlight that this will be the subject of future research, and this manuscript provides a solid foundation. This observation together with the recent advances in deep-learning should therefore facilitate the design of entirely new binding and accessory loops without the need of grafting existing motifs.

Right now, the described potential functionality comes across as a lot of hype with no real proof of concept for reproducing protein-protein molecular recognition properties. What is clear is that this is far from a plug and play strategy for introducing motifs/loops into these scaffolds to mimic functional properties. Under the best of circumstances, prospective practitioners would be faced with fairly sophisticated modeling with little proof of a successful payoff in the end. The hard work of assessing whether this is a viable concept for real world applications is left to the user.

We would like to clarify that our manuscript does not aim to provide a “plug and play” approach for introducing motifs/loops into antibody-like scaffolds for mimicking molecular recognition properties.

Our manuscript presents the structures, binding capacities and potential of a novel two-domain immunoglobulin architecture; which differs from that found in antibodies or nanobodies and, in general, is hard to find in nature. This already represents a substantial advance towards the design of antibody-like antigen-binding proteins with tunable frameworks.

We agree with the Reviewer that protein-protein molecular recognition has not been demonstrated but, as highlighted above and in the manuscript, we have shown that our scaffolds fulfill important requirements for designing protein-protein interactions mediated by loops: robustness for functional loop insertion and structural tunability. Such properties are indeed far from trivial to implement in a protein designed from scratch (these are the first *de novo* protein scaffolds with such properties described to date), and lay the groundwork for subsequent work on designing protein binding interactions. Having said that, high-throughput experimental screening of naive loop libraries (e.g., through yeast surface display) could be readily performed on these scaffolds. We have emphasized this idea in the following fragment of the Discussion section:

The designed binding loops (experimentally tested and/or *in silico* validated) were diverse in terms of sequence, length, amino acid composition and scaffold position. Across three independent β -hairpin sites, the 12-residue EF-hand motif was correctly integrated in loops varying in length (ranging between 15 and 23 residues), and combining amino acids of all types – i.e., polar (S, T, Q, N), charged (K, E, D), aliphatic (A, I, L), aromatic (Y), and turn-forming (G, P) – as shown in Fig. 5b. Such length range is comparable, or larger, than the average length typically found in CDRs of antibodies^{41,42}. Also, amino acid diversity in loop sequences is particularly relevant for designing protein-protein interactions⁴², which require combining polar and hydrophobic interactions, and amino acids controlling loop conformational stability. The designed loops were structurally connected to the rest of the scaffold to a varying degree (Supplementary Fig. 13), which also influences the shape of epitopes that can be targeted. (e.g., protruding loops are in general better suited for targeting concave epitopes, as seen in nanobodies inhibiting enzyme activities⁴). Our results indicate that single-chain Ig dimeric scaffolds have the robustness to accommodate a wide range of loops in multiple sites (both individually and in combination), which holds promise for designing protein binding paratopes in a similar spirit to antibodies or nanobodies. In this regard, this work represents a stepping stone towards *de novo* designing antibody-like protein binding scaffolds. Based on our encouraging results, it will be interesting in future work to design binding loops for protein targets of interest through high-throughput screening of loop libraries^{43,44} (naive or computationally focused) or computational design.

As noted by the Reviewer, the computational approach is sophisticated as it requires combining several techniques for design generation and *in silico* validation. Although this can be a limitation for its broad applicability, it is worth highlighting that this has been key for the high success rate of our designs, which is very relevant to minimize costly experimental efforts. All 9 designs experimentally tested behaved as expected. Among the 5 functional scaffolds, 5 showed native terbium binding affinities, with 4 of them displaying simultaneously two loops differing in terms of sequence length, loop structure and amino acid composition. A paragraph in the Discussion section describes this idea:

From the molecular modeling perspective, the designed single-chain Ig dimer scaffolds fall in a size range (140-160 amino acids) that is quite challenging for *ab initio* structure prediction (without constraints guiding conformational sampling) – the gold-standard *in silico* test for *de novo* designed proteins during the last decade¹⁹. Remarkably, AF2 predicted high-confidence and accurate models for both edge-to-edge and face-to-face, non-functional designs without the need of multiple sequence alignment information, which suggests that the designed sequences strongly encode for their structures. In contrast, the predictions for designs with ligand-binding loops were less confident on the inserted regions, especially those

displaying two loops simultaneously. Yet, by combining protein structure predictions and MD simulations assessing ligand-binding loop structural rigidity, we accurately identified a series of functional designs having structured linkers – e.g. through salt bridge interactions – varying in length and sequence composition that enabled the correct integration of the EF-hand binding motif into our scaffolds. These functional designs turned out to be hyperstable in solution, while showing high-affinity terbium binding across multiple β -hairpin sites. They also showed substantially higher binding affinity (in the low μM range) than a EF-hand terbium loop we recently incorporated in a *de novo* Ig domain¹⁸, likely due to improved predictions both in folding and structural rigidity of the designed loops (Supplementary Fig. 15 and Supplementary Table 5, 6, 7). Overall, the advent of deep-learning structure prediction breaks a historical size limit set by *ab initio* structure prediction, and now allows us to make accurate structural validations for larger proteins than before. As shown here, these methods in combination with physics-based simulation techniques are expected to ease the accurate *de novo* design of increasingly large and complex folds, including multidomain proteins.

Below are specific comments:

1). Considering the stability of the Ig dimer scaffold, it is surprising that the B-factors are so high. This type of thing is expected for membrane proteins, not stable soluble ones. A back of the envelope calculation suggests that the problem may be attributed to the high solvent content of the crystals? The authors should provide this information and comment.

As reasoned by the Reviewer, our crystal structures have a high solvent content and this contributes to increase the overall B-factor: the scIg12 and scIg12+EF3a crystal structures have 68.5% and 69.8% solvent content, respectively. In addition, the protein scaffold surface contains a large number of charged sidechains (lysine, arginine and glutamate). For a fraction of these, there is not enough electron density allowing to trace the rotamer configuration unequivocally, which can also contribute to increasing the B-factors. On Supplementary Table 4 for crystallographic data, we have added the solvent content information and the average B-factors for protein, ligand and water molecules. We have added the following comment at the end of the Methods section “Protein crystallization and structure determination”:

The solvent content of protein crystals was high (68.5% and 69.8% for the scIg12 and scIg12+EF3a structures, respectively), which contributed to increase the B-factors of the solved crystal structures.

2). Particularly disturbing is the high B-factor for the Terbium ion (400 Å²). Basically, this is indicating that the Terbium is misplaced or not there. The anomalous difference density indicates that it is there in the crystal, although the occupancy could be low. Additionally, the position of the Terbium in Fig. 4I (orange sphere) doesn't align well with the anomalous density (at least in the view provided). It would make more sense to place the Terbium into the density and build the loop around that. The coordination geometry of ions incapsulated in EF-Hands is well established (and utilized in the designs of scIg12+EF). Superimposing this structure with the right stereochemistry centered around the ion would provide a good starting point to build the loop. Although local flexibility might still be problematic, given the protein design prowess of the authors, it should be straightforward to predict where this loop has to be using the ion position as a guidepost.

First of all, we would like to distinguish specific from nonspecific terbium binding. After soaking the protein crystals with a terbium chloride solution, terbium ions can either bind to the designed loop (specific binding) or the protein scaffold surface (nonspecific binding), which contains a large number of negatively charged amino acids –i.e., glutamate and aspartate.

We thank the Reviewer for raising the terbium B-factor issue. We have fully revised the scIg12+EF3a crystal structure and found that the terbium ion with a B-factor of 400 Å² (TB 2) should not be there. We erroneously placed the terbium ion in a density corresponding to an adjacent symmetric copy of the protein. We have then removed the ion and performed another round of refinement. In the final refined structure, the three terbium ions correspond to nonspecific binding, not from the designed EF3a loop.

In the crystal structure we have not traced the terbium ion bound to the EF3a loop. We would like to clarify that Fig. 4l displays the terbium ion (orange sphere) in the designed (theoretical) position of our model (in green). For the crystal structure (in gray) we could not trace the backbone of the designed EF3a loop due to the lack of clear electron density, and for this reason it is not displayed in the figure. Also, we have not traced the terbium bound to the loop in the crystal structure as the electron density in that region is not strong enough. Consequently, we can only compare the theoretical position of the terbium ion in the design model with the anomalous density. The right inset shows that both partially overlap, indicating that the terbium ion is mobile and is close to the designed position.

3). Several potential attributes of sc dimers were mentioned, but there are obvious shortcomings, as well if there are thoughts of “replacing” antibodies for clinical applications- half life, immunogenicity, cross reactivity, developability and so forth. Sure, some of this can be engineered into the molecules, but at a cost of time and efficiency.

In summary, the manuscript describes what appears to be a successful design/engineering endeavor to fabricate a family of highly stable β -sheet structures. However, the premise that these structures can be exploited in a practical way remains to be proven. It is essential to actually show that paratopes grafted into these structures perform in a similar fashion as the loops of the protein they are derived from.

We agree with the Reviewer that for therapeutic applications these scaffolds will need to face many challenges. This is common to all novel protein-based therapeutics, but nowadays there exists a broad range of engineering approaches to tackle each of the challenges mentioned by the Reviewer. It is also encouraging that in recent years *de novo* designed proteins with structures unrelated to those described in this manuscript have been successful for some therapeutic applications. We would like to highlight that the potential applications for these structures are not only in therapeutics. Scaffolds binding protein targets of interest could also be used for diagnostic purposes or as affinity reagents in biological research in general; which have lower demands in terms of development for real world applications.

We could not agree more with the Reviewer that our single-chain dimers are a stepping stone towards designing antibody-like protein binding scaffolds. We envision that binding paratopes would be designed, *de novo*, into these scaffolds, rather than grafting existing binding loops as CDRs given the framework differences between antibodies Fv's and our scaffolds. We have emphasized this idea in the following fragment of the Discussion section:

Our results indicate that single-chain Ig dimeric scaffolds have the robustness to accommodate a wide range of loops in multiple sites (both individually and in combination), which holds promise for designing protein binding paratopes in a similar spirit to antibodies or nanobodies. In this regard, this work represents a stepping stone towards *de novo* designing antibody-like protein binding scaffolds. Based on our encouraging results, it will be interesting in future work to design binding loops for protein targets of interest through high-throughput screening of loop libraries^{43,44} (naive or computationally focused) or

computational design. Indeed, recent advances in deep-learning should facilitate the design of entirely new binding loops⁴⁵, without the need of grafting existing motifs and solely based on the target epitope structure.

We take the opportunity to thank the Reviewer again for a valuable contribution.

Reviewers' Comments:

Reviewer #3:

Remarks to the Author:

I have read over the authors responses to my questions. I had no issues with the work and description of the principal design and characterization of the dimer Ig scaffolds. It seems like solid work, which was confirmed by the other two referees. No doubt, sequence diversity can be inserted in the loops of the scaffold without detrimentally affecting the structure and stability of the core the molecules. I remain confused about the rationale for why density for the terbium binding EF-hand loops was completely uninterpretable. This might be a minor point in the context of all the design work, but I'm not quite convinced that more structural information couldn't have been obtained. It is absolutely true that loops with flexibility are invisible in time-resolved experiments like crystallography. However, the anomalous signal from terbium associated with the EF-hand loop suggests that this loop is not totally disordered. The peak is asymmetric showing some disorder, but it appears that there may be enough order that it might be feasible to cassette in the structure of the loop sequence based on known geometries for ion binding in EF-hand motifs. The point being, even with a low resolution experimental model, the rigidity of the terbium-loaded motif would provide information about how well the computer modeling is able to predict the structure of the loop hinge points- which is a big deal for anyone wanting to exploit the scaffolds for their own purposes. All this said, it might be absolutely correct that the loop is untraceable; however, the authors provide no information about the methods they used to try to extract it out. I assume that simulating annealing omit maps were used, but it is surprising that some density from the terbium was not observable, given its electron density and that it was seen in the anomalous difference maps. Also, three non-specific terbium ions were found. Were they observable in the anomalous maps, as well? The point being, for a crystallographer, it doesn't cut it to simply say that the loop was too flexible to trace without giving a description of what was tried. The gory details don't have to be included, but it is important to sketch out what was tried. Overall, this is a strong protein design paper, but softens up when it starts speculating about the potential practical utility of these scaffolds to perform sophisticated functions. nevertheless, the majority of readers would most likely be interested in the design aspects of the paper, the scaffold's utility to be engineered to perform high level molecular recognition functions would probably not take precedence. Those with a background in protein and antibody engineering will have a pretty good idea about the practical feasibility and limitations of exploiting different aspects of the loop insertions into these scaffolds.

Changes made in response to the comments of the Reviewer

In the following section, the comments of the Reviewer are written in italics. Our responses to each of the items are presented under the comments in blue font. Excerpts from the text are written in smaller blue font and indented.

Reviewer 3

I have read over the authors responses to my questions. I had no issues with the work and description of the principal design and characterization of the dimer Ig scaffolds. It seems like solid work, which was confirmed by the other two referees. No doubt, sequence diversity can be inserted in the loops of the scaffold without detrimentally affecting the structure and stability of the core the molecules.

We thank the Reviewer for these positive comments.

I remain confused about the rationale for why density for the terbium binding EF-hand loops was completely uninterpretable. This might be a minor point in the context of all the design work, but I'm not quite convinced that more structural information couldn't have been obtained. It is absolutely true that loops with flexibility are invisible in time-resolved experiments like crystallography. However, the anomalous signal from terbium associated with the EF-hand loop suggests that this loop is not totally disordered. The peak is asymmetric showing some disorder, but it appears that there may be enough order that it might be feasible to cassette in the structure of the loop sequence based on known geometries for ion binding in EF-hand motifs. The point being, even with a low resolution experimental model, the rigidity of the terbium-loaded motif would provide information about how well the computer modeling is able to predict the structure of the loop hinge points- which is a big deal for anyone wanting to exploit the scaffolds for their own purposes. All this said, it might be absolutely correct that the loop is untraceable; however, the authors provide no information about the methods they used to try to extract it out. I assume that simulating annealing omit maps were used, but it is surprising that some density from the terbium was not observable, given its electron density and that it was seen in the anomalous difference maps. Also, three non-specific terbium ions were found. Were they observable in the anomalous maps, as well? The point being, for a crystallographer, it doesn't cut it to simply say that the loop was too flexible to trace without giving a description of what was tried. The gory details don't have to be included, but it is important to sketch out what was tried.

We have tried to trace the EF-hand loop in the crystal structure by different approaches, but without success. When refining the structure without the loop, the difference map does not show uninterpreted electron density in that area. Figure 1a shows a screenshot from the refined structure, where there is no density allowing to trace the loop backbone but some density corresponding to a terbium ion, as supported by the anomalous map (Figure 1b). It is worth noting that anomalous density for other terbium ions binding in a non-specific manner to other sites show larger density, which may indicate again relatively higher mobility in the terbium ion bound to the loop (Fig 1c). Further attempts to include all (or some) loop residues as in the designed conformation worsened the density maps and the overall statistics (Fig. 1d) –e.g., building the loop increases $R_{\text{work}}/R_{\text{free}}$ from 0.210/0.256 to 0.222/0.272. We also generated composite omit maps using simulated annealing, but practically no density was found in that area. Taken together, the difficulty in tracing the loop backbone indicates relatively high flexibility in this area. For this reason, we concluded that it would be more appropriate to present the structure without the loop, instead of overinterpreting the data using the computational model.

Figure 1. Refinement of the EF-hand loop in the scIg12+EF3a structure. **a**, Refined structure without the EF-hand loop. **b**, Refined structure (in yellow) with the anomalous density map and superimposed to the design model (in green). A yellow density blob is found in the loop area and closely matches the designed ion position (white cross). **c**, Another view of the refined structure with the anomalous density map, as in **(b)**, showing large density of terbium ions binding in a non-specific manner (as a result of crystal soaking) to the protein scaffold. **d**, The difference density map of a refined structure including the designed loop residues indicates negative density in the loop. These images were generated with Coot.

We have now provided more details on our attempts to trace the loop in the X-ray crystallography subsection of the Methods section:

To refine the structure of scIg12+EF3a, different approaches were taken to trace the EF-hand loop without success. Without the loop, the difference map did not show uninterpreted electron density in that area, except that corresponding to a terbium ion that was confirmed by the anomalous density map (and was found close to the theoretical design position; Fig 4I). Attempts to manually build the EF-hand loop backbone based on the design model led to worse density maps and refinement statistics. We also generated composite omit maps using simulated annealing, but no density was found in that area either. Overall, the challenge of tracing the loop backbone suggests relatively high flexibility, and hence we concluded it is more appropriate to display the structure without the loop.

Overall, this is a strong protein design paper, but softens up when it starts speculating about the potential practical utility of these scaffolds to perform sophisticated functions. nevertheless, the majority of readers would most likely be interested in the design aspects of the paper, the scaffold's

utility to be engineered to perform high level molecular recognition functions would probably not take precedence. Those with a background in protein and antibody engineering will have a pretty good idea about the practical feasibility and limitations of exploiting different aspects of the loop insertions into these scaffolds.

We thank the Reviewer for these positive comments. As indicated in the Discussion section, this work represents a stepping stone towards the de novo design of antibody-like protein binding scaffolds. This lays the groundwork for future work on the computational design of new loops or high-throughput screening of loop libraries on these scaffolds.

We take the opportunity to thank the Reviewer again for a valuable contribution.